# Left hemisphere dominance for bilateral kinematic encoding in the human brain

Christina M Merrick[1]*, Tanner C Dixon[2], Assaf Breska[1], Jack Lin[3], Edward F Chang[4], David King-Stephens[5], Kenneth D Laxer[5], Peter B Weber[5], Jose Carmena[2,6,7], Robert Thomas Knight[1,2,4,7], Richard B Ivry[1,2,7]

[1]Department of Psychology, University of California, Berkeley, Berkeley, United States; [2]UC Berkeley – UCSF Graduate Program in Bioengineering, University of California, Berkeley, Berkeley, United States; [3]Department of Neurology, University of California at Irvine, Irvine, United States; [4]Department of Neurological Surgery, University of California San Francisco, San Francisco, San Francisco, United States; [5]Department of Neurology and Neurosurgery, California Pacific Medical Center, San Francisco, United States; [6]Department of Electrical Engineering and Computer Sciences, University of California, Berkeley, Berkeley, United States; [7]Helen Wills Neuroscience Institute, University of California, Berkeley, Berkeley, United States

**Abstract** Neurophysiological studies in humans and nonhuman primates have revealed movement representations in both the contralateral and ipsilateral hemispheres. Inspired by clinical observations, we ask if this bilateral representation differs for the left and right hemispheres. Electrocorticography was recorded in human participants during an instructed-delay reaching task, with movements produced with either the contralateral or ipsilateral arm. Using a cross-validated kinematic encoding model, we found stronger bilateral encoding in the left hemisphere, an effect that was present during preparation and was amplified during execution. Consistent with this asymmetry, we also observed better across-arm generalization in the left hemisphere, indicating similar neural representations for right and left arm movements. Notably, these left hemisphere electrodes were centered over premotor and parietal regions. The more extensive bilateral encoding in the left hemisphere adds a new perspective to the pervasive neuropsychological finding that the left hemisphere plays a dominant role in praxis.

*For correspondence: cmerrick@berkeley.edu

## Editor's evaluation

Based on a rare set of human intracranial recordings, this paper by Merrick and colleagues asks how the neural processes that generate arm movements are distributed across cortical hemispheres. The authors demonstrate that these contributions are not equal. The key result is that the left hemisphere shows stronger bilateral representations than the right hemisphere. This effect is present during movement preparation but is further accentuated during movement execution. Taken together, the findings provide important new insight into the hemispheric asymmetries that underlie manual actions.

## Introduction

A primary tenet of neurology is the contralateral organization of movement. The vast majority of the fibers from the corticospinal tract cross to the opposite side of the body (**Nyberg-Hansen and Rinvik, 1963**) and functionally, hemiparesis resulting from cortical stroke is manifest on the contralateral side of the body (**Bourbonnais and Vanden Noven, 1989**). Although direct control of arm movements

**eLife digest** The brain is split into two hemispheres, each playing the leading role in coordinating movement for the opposite side of the body: lesions on the left hemisphere therefore often result in difficulties moving the right arm or leg, and vice versa. In fact, very few anatomical connections exist between a given hemisphere and the body parts on the same (or 'ipsilateral') side. Yet, movements produced with only one limb still engage both sides of the brain, with the hemisphere which does not control the action production, still encoding the direction and speed of the movement. Previous evidence also indicate that the two hemispheres may not have equal roles when coordinating ipsilateral movements.

Merrick et al. aimed to shed light on these processes; to do so, they measured electrical activity from the surface of the brain of six patients as they moved their arms to reach a screen. The results revealed that, while the right hemisphere only encoded information about the opposite arm, the left hemisphere contained information about both arms. Finer analyses showed that, for both hemispheres, moving the opposite arm was strongly associated with activity in the primary motor cortex, a region which helps to execute movements. However, in the left hemisphere, movements from the ipsilateral arm were related to activity in brain areas involved in planning and integrating different types of sensory information.

These findings contribute to a better understanding of how the motor system works, which could ultimately help with the development of brain-machine interfaces for patients who need a neuroprosthetic limb.

is primarily mediated through contralateral projections, unimanual arm movements elicit bilateral activity in the primary motor cortex (M1, *Babiloni et al., 1999*; *Ghacibeh et al., 2007*), indicating that neural activity in the ipsilateral hemisphere contains information relevant to ongoing movement. Correspondingly, kinematic and movement parameters of the ipsilateral limb can be decoded from ipsilateral hemisphere intracortical recordings in monkeys (*Ganguly et al., 2009*; *Ames and Churchland, 2019*) and from electrocorticography (ECoG) in humans (*Bundy et al., 2018*; *Ganguly et al., 2009*; *Wisneski et al., 2008*). Ipsilateral signals represent an intriguing source of neural activity, both for understanding how activity across the two hemispheres results in coordinated movement and because this information might be exploited for rehabilitative purposes.

While it is established that information about unimanual movements is contained within the ipsilateral hemisphere, there remains considerable debate about what this signal represents. Previous studies have centered on the question of whether ipsilateral representations overlap or are independent of contralateral representations, leading to mixed results. Consistent with the overlap hypothesis, neural activity for the contralateral and ipsilateral limb movements shows several similarities, including shared target tuning preferences and the ability to cross predict kinematic features from a model trained on the opposite arm (*Bundy et al., 2018*; *Cisek et al., 2003*; *Steinberg et al., 2002*; *Willett et al., 2020*). Consistent with the independence hypothesis, intracortical recordings in monkeys have revealed that the lower dimensional representations of the two arms lie in orthogonal subspaces (*Ames and Churchland, 2019*; *Heming et al., 2019*). These hypotheses are not mutually exclusive. For example, the degree of overlap or independence may depend on the gesture type (e.g., overlapping representations for grasping but not arm movement, *Downey et al., 2020*), or brain region (e.g., premotor cortex displays stronger preservation of tuning preferences across the two arms than primary motor cortex, *Cisek et al., 2003*).

One factor that has received little attention in this literature is the recording hemisphere. This is surprising given the marked asymmetries between the two hemispheres in terms of praxis (*Corballis et al., 2012*; *Rothi et al., 1997*). Tracing back to the early 20th century, marked hemispheric asymmetries have been defined by the behavioral deficits observed following unilateral brain injury (*Schaefer et al., 2007*; *Liepmann, 1908*, cited in *De Renzi and Lucchelli, 1988*). Apraxia, an impairment in the production of coordinated, meaningful movement in the absence of muscle recruitment deficits, is much more common after left compared to right hemisphere insult (*Haaland et al., 2000*; *De Renzi and Lucchelli, 1988*). Moreover, left hemisphere stroke will frequently result in apraxic symptoms for gestures produced with either hand, as well as impairments in action comprehension (*De Renzi and*

*Lucchelli, 1988*). Hemispheric asymmetries are also evident in neuroimaging activation patterns in healthy participants, with the left hemisphere having stronger activation during ipsilateral movement than the right hemisphere, especially with increasing task difficulty (*Chettouf et al., 2020*; *Verstynen et al., 2005*; *Verstynen and Ivry, 2011*; *Schäfer et al., 2012*). These patterns raise the possibility that the ipsilateral cortical representation differs between the left and right hemispheres.

In the present study, we use intracranial recordings from the cortical surface (ECoG) to examine the degree of cortical overlap for ipsilateral and contralateral upper limb movement in the left and right hemispheres. The data were collected from six patients, three with left hemisphere implants and three with right hemisphere implants, while they engaged in an instructed-delay reaching task. We focus on predicting the temporal dynamics of high-frequency activity (HFA; 70–200 Hz), a surrogate for infra-granular single-unit activity and supragranular dendritic potentials (*Leszczyński et al., 2020*), which tracks local activation of the cortex (*Muthukumaraswamy, 2010*). Going beyond previous studies that use decoding models which combine multiple neural features from multiple electrodes to predict kinematics, we employed an encoding model which uses kinematic features to predict neural activity for each electrode, allowing us to retain the high spatial and temporal resolution of the ECoG signal. This approach allows us to create high-resolution topographic maps depicting encoding strength on the surface of the cortex for movements produced with the contralateral and ipsilateral arms. This is preferable to projecting the weights obtained from decoding models since these models have difficulty disambiguating between informative and uninformative electrodes (*Kriegeskorte and Douglas, 2019*). Moreover, our approach provides a way to map kinematics to neural activity in a time-resolved manner (rather than as single weights), allowing us to identify time ranges of representational overlap and divergence across the two arms for each electrode.

## Results

### Behavior

We used a delayed response, out-and-back reaching task. On each trial, a cue indicating the target location was presented on a touchscreen followed, after a short delay, by an imperative signal. Participants were instructed to prepare to move to the target during the delay period. The participant was free to move at their own pace, with the instructions emphasizing that the participant should focus on touching the screen near the target and then returning to the start position. Left and right arm reaches were tested in separate blocks, with the position of the nonresponding hand fixed throughout the block.

*Table 1* summarizes the total number of successful trials, along with the reaction time and movement time data. A trial was considered unsuccessful if the reach was initiated before the go cue or if contact with the touchscreen was outside the boundary of the target. The percentage of unsuccessful trials was low, ranging between 0% and 12.5% across individuals. The movements had roughly, bell-shaped velocity profiles for the outbound and the inbound segments (*Figure 1C, E*) and the outbound reaches were, on average, faster than inbound reaches. The marked interindividual differences in reaction time and movement time reflect the fact that the instructions emphasized accuracy and smoothness.

**Table 1.** Summary of performance measures for each participant.
Reaction times (RT) are averages over left and right arm reaches since RT differences were negligible for the two arms.

| Patient ID | Age | Handedness | RT | Outbound reach | Inbound reach | Total trials |
|---|---|---|---|---|---|---|
| L1 | 36 | Right | 392 (102) | 703 (118) | 907 (287) | 152 |
| L2 | 26 | Right | 1574 (871) | 598 (173) | 1003 (406) | 146 |
| L3 | 55 | Ambidextrous | 771 (345) | 946 (183) | 1429 (252) | 148 |
| R1 | 26 | Right | 518 (194) | 940 (211) | 1027 (345) | 132 |
| R2 | 32 | Right | 335 (55) | 602 (94) | 590 (120) | 145 |
| R3 | 34 | Right | 534 (157) | 721 (145) | 1017 (318) | 145 |

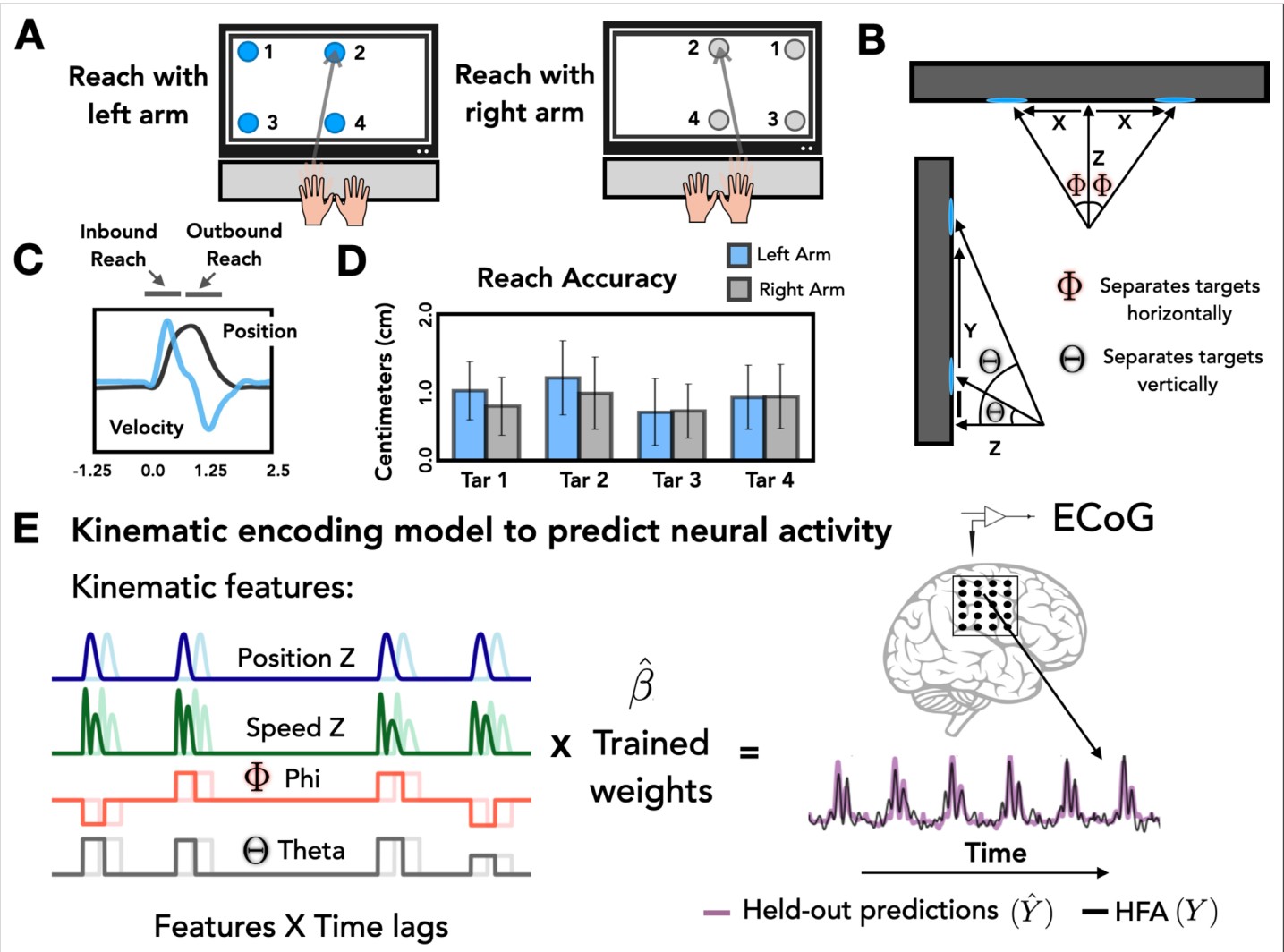

**Figure 1.** Task and model design. (**A**) Task overview. Participants performed an instructed-delay reaching task, moving to targets that appeared on a touchscreen monitor with either the left or right arm. (**B**) Target positions. The position of each target with respect to the start position of the reaching arm can be defined on the basis of three Cartesian coordinates (*X*, *Y*, *Z*) and two spherical angles (theta and phi). (**C**) Representative kinematics. Average estimated position and velocity traces for a representative series of trials performed with the left arm. (**D**) Reach accuracy. Accuracy was quantified as the absolute distance from the center of each target (target diameter = 2.5 cm) to the touch location for all four targets with the left (blue) and right (gray) arm. (**E**) Kinematic encoding model. Time lagged estimated kinematic features were used to the predict high-frequency activity (HFA) for each electrode using ridge regression. Four kinematic features were included in the model: Position in the *Z* dimension, speed in the *Z* dimension, and the two spherical angles phi and theta. Kinematic features were trained on a subset of the HFA data and predictions of HFA were evaluated on held-out test sets.

The online version of this article includes the following figure supplement(s) for figure 1:

**Figure supplement 1.** Actual vs. estimated kinematics.

At a more fine-grained level of spatial accuracy, we calculated the distance from the center of each target to the touch location for each trial. On average, the mean distance from the center of the 2.5 cm circle was 0.80 cm (SD = 0.10 cm) for right-handed reaches and 0.90 cm (SD = 0.17 cm) for left-handed reaches (*Figure 1D*). These values did not differ from one another [$t = 1.538$, $p = 0.22$].

### Stronger bilateral encoding in the left hemisphere

We examined the extent to which movement kinematics were encoded for contralateral and ipsilateral reaches in individual electrodes. To do this we fit a kinematic encoding model that maps continuous kinematic features to the HFA signal (*Figure 1E*) for the 665 electrodes meeting our inclusion criteria.

This procedure was done separately for contralateral and ipsilateral reaches. We quantified the cross-validated model fit by generating HFA predictions using the kinematic features from held-out trials of the same condition and calculating prediction performance as the square of the linear correlation ($R^2$) between the predicted and actual HFA signal (*Figure 2B*).

*Figure 2A* displays $R^2$ values for each electrode for the contralateral and ipsilateral conditions, presented on the individual patient MRIs. Electrodes with high prediction performance were primarily located in arm areas of sensorimotor cortex. In line with previous research (*Downey et al., 2020*), a sizeable percentage of the electrodes were able to predict the HFA at or above our criterion of $R^2 >$ 0.05 (examples shown in *Figure 2B*). This degree of prediction was observed not only when the data were restricted to contralateral movement (31% of electrodes), but also when the data were from ipsilateral movement (25%). A number of electrodes (24%) were predictive in both the contralateral and ipsilateral models. Electrodes that did not meet this criterion for either arm are represented as small dots in *Figure 2A* and were excluded from further analysis, leaving a total of 216 predictive electrodes (32%, 141 = left hemisphere, 75 = right hemisphere).

Before comparing prediction performance across the two hemispheres, we first evaluated the distribution of the electrodes in the right and left hemispheres. Electrode placement for ECoG is determined solely for clinical purposes; as such, hemispheric asymmetries could arise from differences in coverage rather than functional differences. To evaluate coverage, we categorized the position of all electrodes based on a cortical parcellation (*Desikan et al., 2006*; *Figure 2—figure supplement 1*). When the categorization data were pooled across the three left hemisphere and the three right hemisphere patients, the proportion of electrodes was similar across the eight parcellations that encompass premotor, sensorimotor motor, and parietal regions [averages: $premotor_{right}$ = 11%, $premotor_{left}$ = 10%, $sensorimotor_{right}$ = 17%, $sensorimotor_{left}$ = 16%, $parietal_{right}$ = 5%, $parietal_{left}$ = 3%; $\chi^2(7)$ = 0.057, p = 0.99; *Figure 2—figure supplement 1*].

Given the similar distributions, we next asked whether prediction performance for the two arms differed across the two hemispheres. *Figure 2C* compares the predictive performance of each electrode for the contralateral and ipsilateral conditions for patients with left or right hemisphere grids. Values close to the unity line yield similar predictions for the two conditions; values off the unity line indicate that encoding is stronger for one arm compared to the other. To compare prediction performance at the group level, we fit a permutation-based mixed-effects model with fixed factors of *Arm* and *Hemisphere* and a random factor of *Participant*. We found a main effect of *Arm* with contralateral reaches being encoded more strongly than ipsilateral reaches [ $\chi^2(1)$ = 29.34, p < 0.001]. We found no effect of *Hemisphere* [ $\chi^2(1)$ = 0.46, p > 0.50], but we found a significant interaction between *Arm* and *Hemisphere* [ $\chi^2(1)$ = 12.03, p < 0.001].

To further explore this interaction, we calculated the difference between the $R^2$ values for the contralateral and ipsilateral conditions for each electrode, using this as a proxy of an encoding bias between the two arms (*Figure 2C*, upper right corner of each scatterplot). Values close to zero indicate similar encoding across the two arms, whereas positive values correspond to stronger contralateral encoding and negative values stronger ipsilateral encoding. The distribution for each condition was positively skewed indicating that, overall, there was a bias for better encoding for contralateral reaches [permutation test: $p_{right}$ < 0.001, $p_{left}$ < 0.001]. However, there was a significant difference in the distributions for the two hemispheres: The bias scores were lower in the left hemisphere compared to the right hemisphere [permutation test: p < 0.001] indicating stronger bilateral encoding in the left hemisphere. We also found that the contralateral bias becomes weaker the further the electrodes are from the primary motor cortex, an effect observed in both hemispheres [$r_{left}$ = −0.48, $p_{left}$ < 0.001, $r_{right}$ = −0.45, $p_{right}$ < 0.001; *Figure 2—figure supplement 3*].

## Opposing patterns of kinematic encoding for the left and right hemispheres during planning and execution

As neural activity unfolds from preparation to movement, the underlying computations may change substantially (*Elsayed et al., 2016*). To examine if hemispheric asymmetries in encoding depend on task state, we repeated the mixed-effects model described in the previous section, but now added a factor *Task Phase*, separating the data to test the held-out predictions during the instruction and movement phases (*Figure 3A*). The effect of *Arm* was again significant, with contralateral reaches more strongly encoded than ipsilateral reaches [$\chi^2(1)$ = 16.19, p < 0.001]. The main effects of *Hemisphere* [

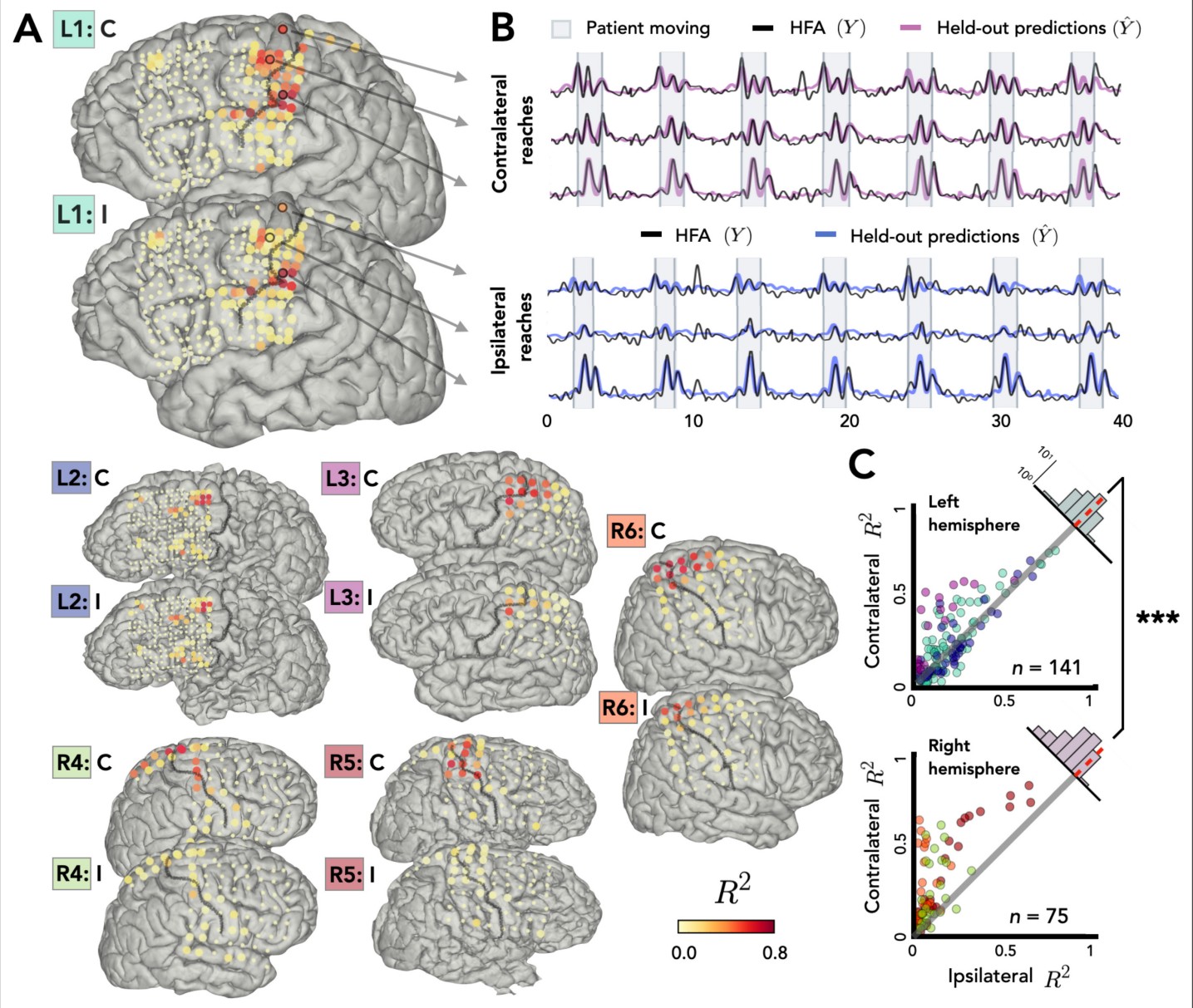

**Figure 2.** Stronger bilateral encoding in left hemisphere. Held-out prediction performance ($R^2$) was computed for each electrode during contralateral reaches (C) and ipsilateral reaches (I). $R^2$ was calculated as the squared linear correlation between the actual high-frequency activity (HFA) and the predictions based on the model. (**A**) Prediction performance maps for individual patients. Performance of each electrode, shown at the idiosyncratic electrode location for each participant (location based on clinical criteria). Electrodes that did not account for at least 0.05% of the variance ($R^2 < 0.05$) in either the contralateral or ipsilateral condition are shown as smaller dots. (**B**) Model predictions. Representative time series of the actual HFA and model-based predictions for three electrodes during contralateral and ipsilateral reaches. (**C**) Summary across patients. Scatter plot displaying $R^2$ values separately for patients with electrodes in either the left (upper) or right (lower) hemisphere. Individual patients can be identified in the scatterplot by the color surrounding the patient label (e.g., L1) in A. $R^2$ for contralateral predictions are plotted against $R^2$ for ipsilateral predictions. Electrodes close to the unity line encode both arms equally whereas electrodes off the unity line indicate stronger encoding of one arm. Points above the unity line indicate stronger encoding of the contralateral arm. These differences are summarized in the frequency histograms in the upper right of each panel. The histogram shows less of a shift in the left hemisphere, a signature consistent with stronger bilateral encoding. ***p < 0.001, permutation test.

The online version of this article includes the following figure supplement(s) for figure 2:

**Figure supplement 1.** Similar distribution in left and right hemispheres of electrodes in premotor, sensorimotor, and parietal regions.

**Figure supplement 2.** Assessing explainable variance in the encoding model.

**Figure supplement 3.** Distance from central sulcus negatively correlates with bilateral encoding.

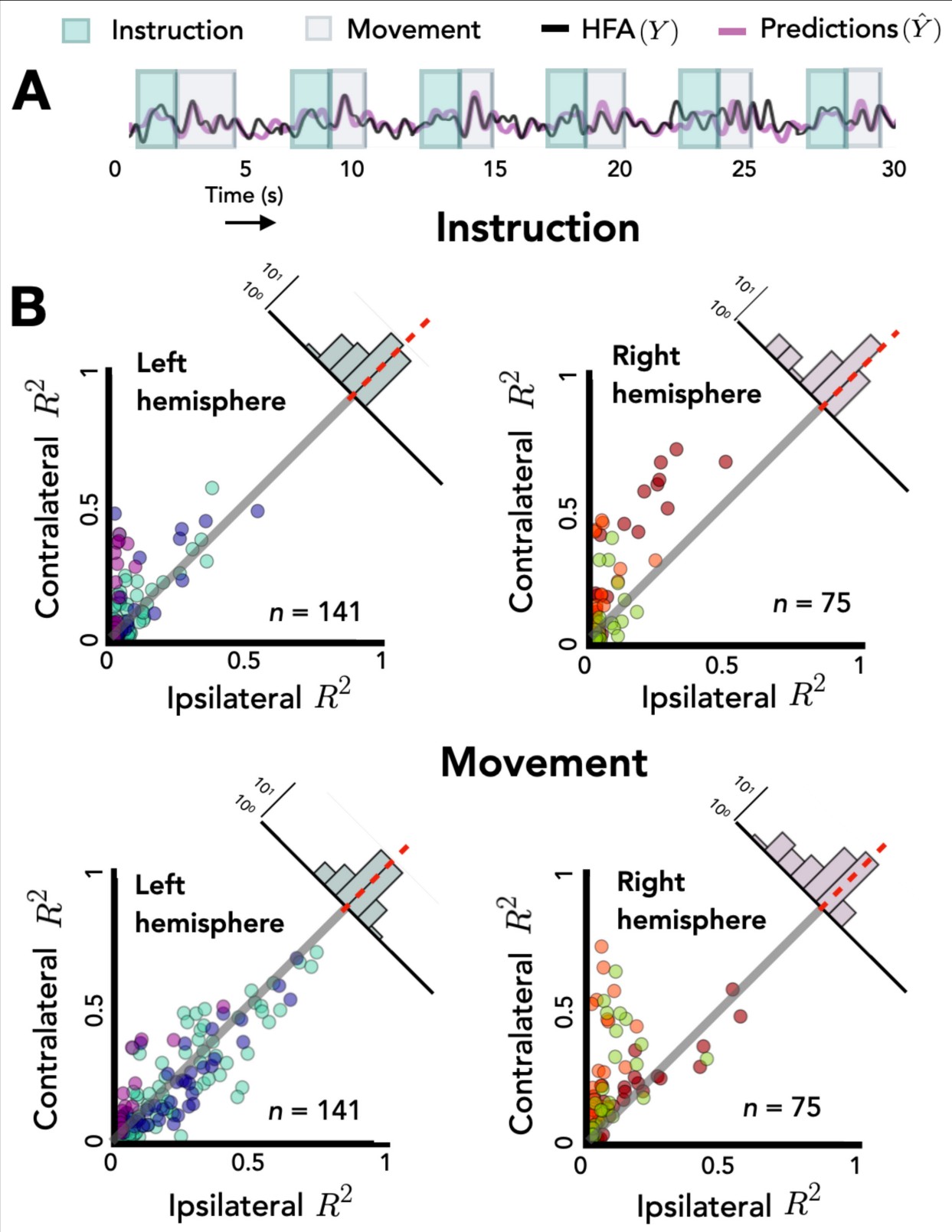

**Figure 3.** Opposing encoding patterns for left and right hemispheres across task phase. For all predictive electrodes the time series was segmented into instruction and movement epochs. $R^2$ was then calculated separately for each epoch. (**A**) Example model predictions. Time series of a representative electrode with boxes surrounding the instruction (teal) and movement epochs (gray). (**B**) Prediction performance during movement and instruction. Comparison of $R^2$ values for contralateral and ipsilateral predictions during the instruction epoch (top) and the movement epoch (bottom)

*Figure 3 continued*

for patients with electrodes in the left hemisphere (left) or right hemisphere (right). Bilateral encoding was stronger in the left hemisphere, an effect that was especially pronounced during the movement phase.

---

$\chi^2(1) = 0.72$, p > 0.40] and *Task Phase* were not significant [$\chi^2(1) = 0.01$, p > 0.90]. Importantly, there was a three-way interaction between *Arm*, *Hemisphere*, and *Task Phase* indicating that the level of encoding for the two arms varied across the two hemispheres for the two task phases [$\chi^2(4) = 22.47$, p < 0.001].

To explore this interaction we again examined the distribution of difference scores for each electrode. The right hemisphere electrodes show a positive skew in both the planning and movement phase [permutation test: $p_{right\_move} < 0.001$, $p_{right\_planning} < 0.001$]. However, this pattern is only seen in the left hemisphere during the planning phase [$p_{left\_planning} < 0.001$]; the mean difference score was not statistically different from zero for the left hemisphere electrodes in the movement phase [$p_{left\_move} = 0.482$]. Analyzing simple effects within each hemisphere, we found that the difference score was smaller (i.e., more bilateral encoding) in the left hemisphere during the movement phase compared to the planning phase [$R^2_{left\_move} = 0.01$, $R^2_{left\_planning} = 0.04$, p < 0.001]. In contrast, the opposite pattern was observed in the right hemisphere, with encoding being more bilateral during the instruction phase [$R^2_{right\_move} = 0.13$, $R^2_{right\_planning} = 0.10$, p < 0.001]. Taken together, these results suggest that the left and right hemispheres may have different roles in bilateral encoding with regard to task phase.

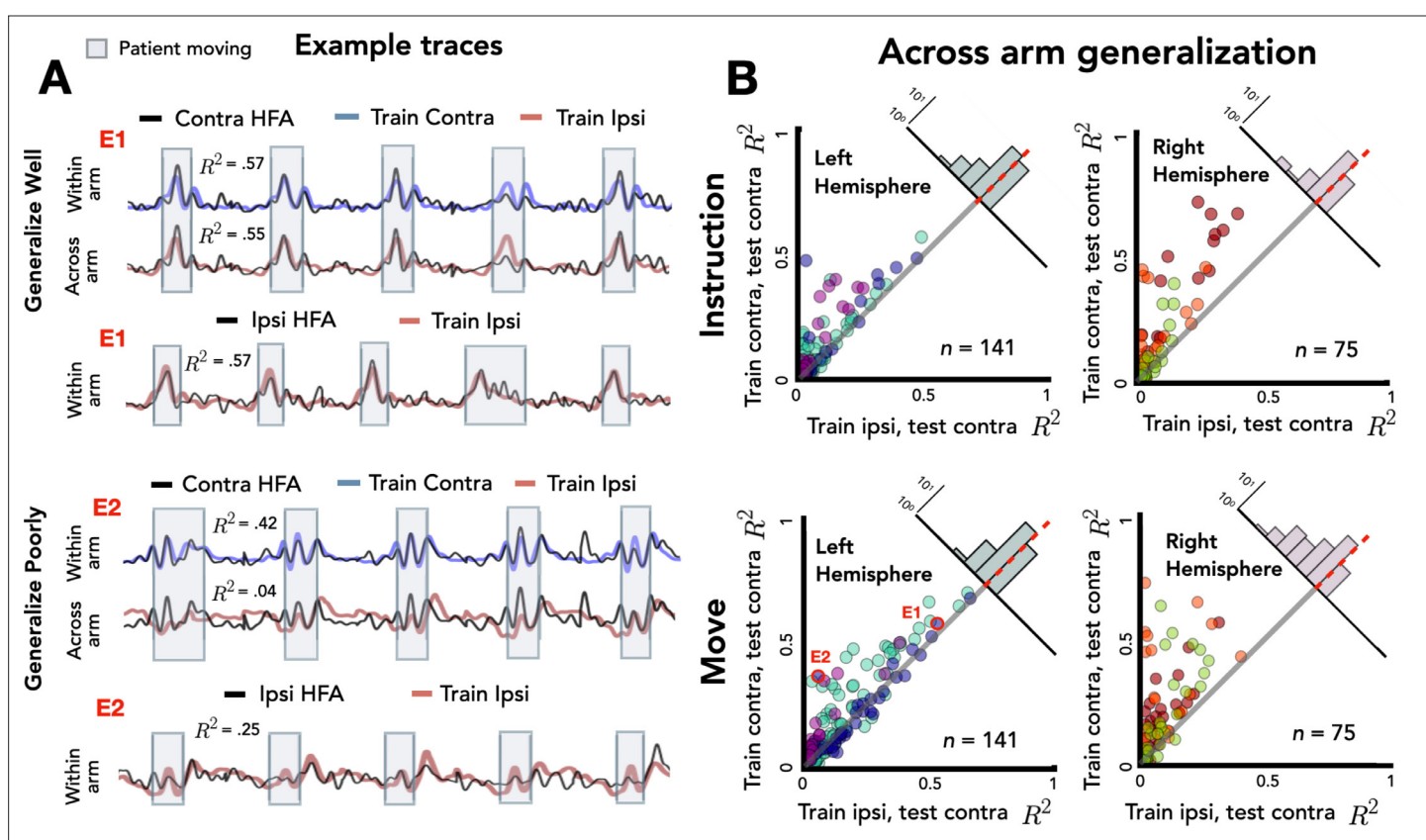

**Figure 4.** Stronger across-arm generalization in the left hemisphere. Across-arm predictions were created by training the model on ipsilateral reaches and using the trained weights to predict high-frequency activity (HFA) during contralateral reaches. (Within-arm predictions generated the same as in *Figures 2 and 3*.) Electrodes close to the unity line have overlapping neural representations across the two arms whereas electrodes off the unity line indicate that the two arms are being encoding differentially. (**A**) Model predictions. Predicted and actual HFA for two electrodes selected from the distribution of left hemisphere electrodes during movement, one that generalizes well across arms (E1) and one that fails to generalize (E2). Bottom row shows within-arm performance for ipsilateral trials, demonstrating that the failure to generalize across arms does not necessarily indicate poor ipsilateral performance. (**B**) Across-arm generalization across patients. $R^2$ for within-arm predictions plotted against $R^2$ for across-arm predictions, with the analysis performed separately for the instruction and movement phases. Left hemisphere electrodes showed better generalization than right hemisphere electrodes, an effect that was magnified in the movement phase.

## Across-arm generalization: more overlap between arms in the left hemisphere

The preceding analyses focused on an encoding analysis for within-arm prediction. We next evaluate the overlap between the neural representations for contralateral and ipsilateral movements. To this end, we examined across-arm prediction performance by training the kinematic encoding model with the data from movements produced with one arm and testing prediction performance using the data from movements produced with the other arm.

*Figure 4A* shows the traces for two representative electrodes, one that shows good generalization across the two arms and the other that shows poor generalization. For the electrode that shows good generalization (E1), prediction performance for held-out contralateral reaches is comparable when the model is trained on data from either the contralateral or ipsilateral arm. This suggests that there is overlap between the neural representations for reaches performed with either upper limb for this electrode. In contrast, the electrode showing poor generalization (E2) showed good prediction for contralateral reaches when trained with contralateral data, but poor prediction when trained with ipsilateral data. Here, the neural representations for the arms do not overlap. Note that E2 showed relatively strong within-arm ipsilateral encoding ($R^2$ = 0.25); thus, the inability of this electrode to generalize across arms is not a result of poor encoding of the ipsilateral arm. Rather, E2 encodes movement produced by either arm, but the manner in which they are encoded differs.

*Figure 4B* summarizes the comparison of within-arm prediction (*y*-axis) against across-arm prediction (*x*-axis), with the data separated for the instruction and movement phases. In this depiction, electrodes close to the unity line have overlapping neural representations during contralateral and ipsilateral movements, whereas electrodes off the unity line encode the two arms differently. We again used a permutation-based mixed-effects model, this time with fixed factors of *Generalization*, *Task Phase*, and *Hemisphere* and a random factor of *Participant*. As expected, we found a main effect of *Generalization* with within-arm predictions outperforming across-arm predictions [$\chi^2(1)$ = 103, p < 0.001]. There was also a main effect of *Task Phase* [$\chi^2(1)$ = 30.51, p < 0.001] with better prediction accuracy during the movement phase compared to the instruction phase. The main effect of *Hemisphere* was not significant [$\chi^2(1)$ < 0.01, p > 0.90], but there was a significant three-way interaction between the three factors [ $\chi^2(4)$ = 19.56, p < 0.001].

To evaluate this interaction, we used a difference score, calculated by subtracting the across-arm $R^2$ from the within-arm $R^2$ for each electrode (*Figure 4B*, upper right corner of each scatterplot). As such, the larger (more positive) the score, the poorer the electrode is in predicting activity when the movement is produced with the other arm. Overall, across-arm generalization was better during the instruction phase compared to the movement phase [$R^2_{instruction}$ = 0.05, $R^2_{movement}$ = 0.08; p < 0.001]. In terms of laterality differences, the left hemisphere showed stronger across-arm generalization (lower difference values) than the right hemisphere [$R^2_{left}$ = 0.04, $R^2_{right}$ = 0.11; p < 0.001]. Analyzing simple effects within each task phase, the left hemisphere had better across-arm generalization compared to the right hemisphere for both the instruction and movement phase [simple effect analysis: $p_{instruct}$ < 0.001, $p_{move}$ < 0.001]. In addition, better across-arm generalization was found during the instruction phase in both the left and right hemispheres [simple effect analysis: $p_{left}$ < 0.001, $p_{right}$ < 0.001]. In sum, the three-way interaction reflects the fact that electrodes in the left hemisphere exhibit better generalization across arms than right hemisphere electrodes, an effect observed in both the instruction and movement phases. In addition, the difference in across-arm generalization between phases is more pronounced in the right hemisphere.

Considered together, the within- and across-arm encoding results indicate that left hemisphere electrodes show stronger bilateral encoding and more similar encoding during left and right arm movements. Interestingly, although overall encoding was stronger during movement, across-arm generalization was stronger during the instruction phase.

## Temporal and spatial topography of across-arm generalization

To examine how generalization varied across the cortex, we categorized each electrode as showing either good across-arm generalization (decrease of up to 20% relative to within-arm performance) or poor across-arm generalization (decrease of more than 50%; *Figure 5A*). We focused on the extremes of the generalization distribution based on the assumption that these electrodes were more likely to

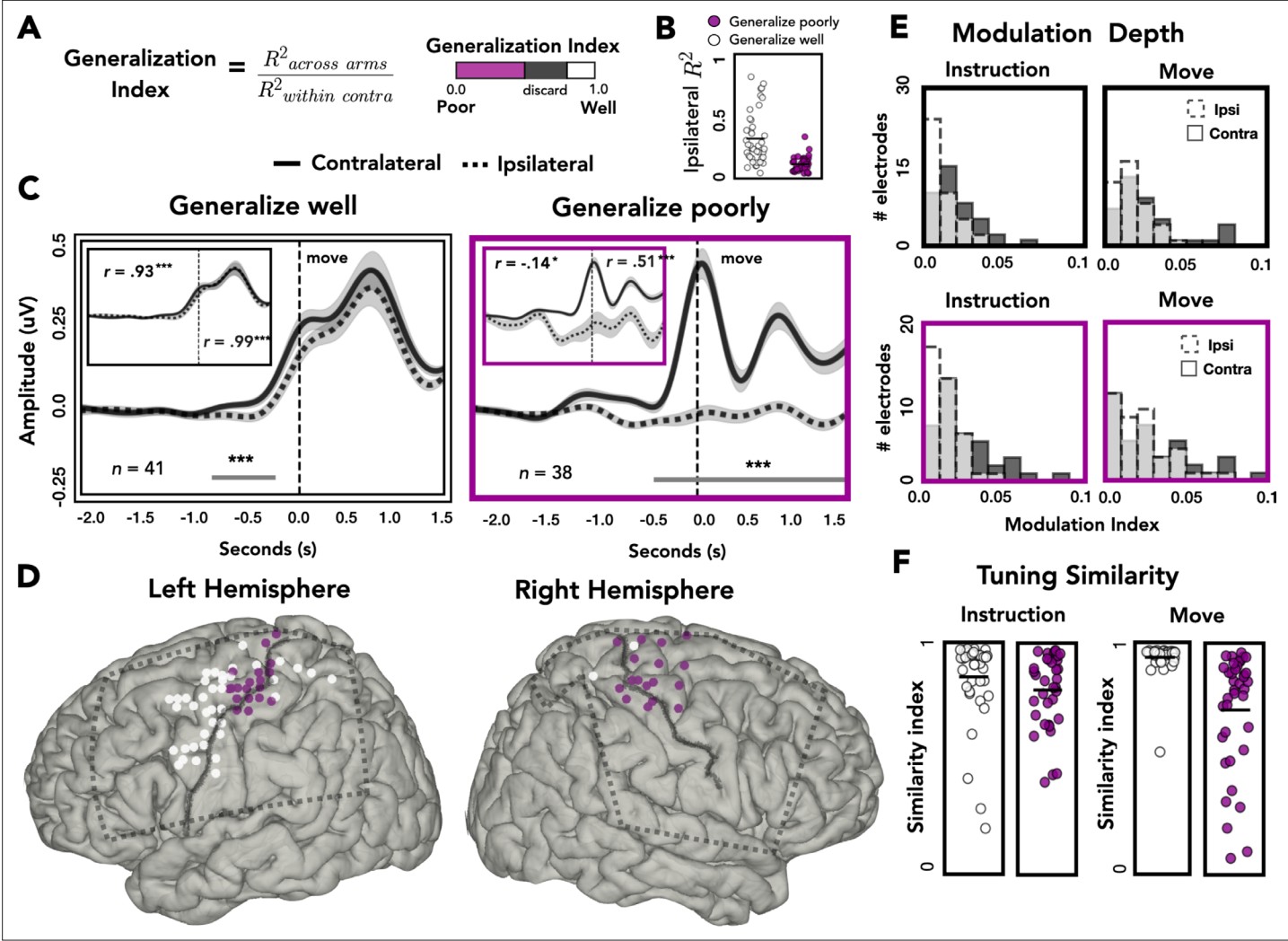

**Figure 5.** Spatial and temporal relationships of across-arm generalization. (**A**) Generalization index. Electrodes were classified as showing good across-arm generalization (white, generalization index >0.80) or poor across-arm generalization (magenta; generalization index <0.50). (**B**) Ipsilateral encoding strength. Ipsilateral encoding strength for electrodes that generalize well and generalize poorly. (**C**) Amplitude differences across arms. Average contralateral (solid line) and ipsilateral (dashed line) high-frequency activity (HFA) for electrodes that generalize well across arms (left) or generalize poorly (right). Significant clusters are represented with a gray line. Inset: same data but standardized to highlight shape of the time series independent of absolute amplitude. (**D**) Spatial distribution of across-arm generalization. Electrodes that generalize well across arms (white) were primarily located in dorsal and ventral premotor regions of the left hemisphere. Electrodes that generalize poorly (magenta) were clustered around the putative arm area of the central sulcus in both the left and right hemispheres. Dashed rectangles indicate extent of electrode coverage across the three patients for each hemisphere. (**E**) Modulation depth. Depth of tuning was calculated during instruction or movement with either the ipsilateral or contralateral hand. Greater modulation was found during contralateral reaches and during movement. (**F**) Tuning similarity. Across arms tuning similarity was calculated for electrodes that generalize well (white) or poorly (magenta). Electrodes that generalize well across arms had significantly more tuning similarity than electrodes that did not generalize. *p < 0.05, ***p < 0.001, cluster permutation test, Pearson's correlation.

The online version of this article includes the following figure supplement(s) for figure 5:

**Figure supplement 1.** Distance from central sulcus positively correlates with degree of overlap between representations for the left and right movements.

share similar underlying neural profiles. This also allowed us to have similar numbers of electrodes in each group.

*Figure 5B* plots ipsilateral encoding strength ($R^2$) for electrodes that either generalize well or generalize poorly. Electrodes that generalize poorly have lower ipsilateral encoding strength than electrodes that generalize well [$t$ = 5.921, $p < 0.001$]. Despite this difference, 61% (23/38) of electrodes that generalize poorly meet our encoding criteria of $R^2 > 0.05$.

To examine the dynamics of representational overlap and divergence, we averaged the time-resolved HFA amplitude across electrodes, restricted to those showing good or poor across-arm generalization. *Figure 5C* displays the average time series for contralateral (solid line) and ipsilateral (dashed line) predictions for electrodes that generalize well (white) or poorly (magenta). The temporal profile of HFA is similar for electrodes that generalize well, showing a single peak in the movement phase. A cluster-based permutation test identified two periods where the HFA amplitude differed for contralateral and ipsilateral reaches, one during instruction and one well into the movement period. In contrast, the temporal profiles are radically different for those that generalize poorly. The lack of correspondence likely arises, at least in part, because of the weak modulation for these electrodes during ipsilateral reaches.

It is possible that similarity in temporal structure is obscured in the preceding analysis by the differences in HFA amplitude for the electrodes that showed poor generalization. To control for this, we standardized the time series data by dividing each sample by the overall standard deviation (insets: *Figure 5C*). Using the standardized traces, we calculated the linear correlation coefficient between the contralateral and ipsilateral traces, separately for instruction and movement. As expected, electrodes that generalized well across arms showed strong across-arm correlations for both task phases (inset: *Figure 5C*, left). In contrast, for electrodes that generalize poorly across arms, the correlation between arms was negative during instruction and then rose to a moderate positive correlation during movement (inset: *Figure 5C*, right). Thus, the poor generalization of these electrodes is due in part to the temporal divergence of the two arms during instruction, where the ipsilateral trace becomes inhibited compared to the contralateral trace. Interestingly, although the ipsilateral trace remains inhibited during movement, the temporal structure between the two arms reemerges.

We next examined the relationship between generalization, hemisphere and spatial position on the cortical surface. As can be seen in *Figure 5D*, electrodes that generalize well were predominantly found in the left hemisphere (proportion of electrodes that generalize across patients: L1 = 22%, L2 = 39%, L3 = 29%, R1 = 0%, R2 = 5%, R3 = 4%; $\chi^2(2)$ = 48813, p < 0.001). In contrast, electrodes showing poor generalization were observed in both hemispheres (proportion of electrodes that generalize poorly: L1 = 21%, L2 = 5%, L3 = 6%, R1 = 14%, R2 = 15%, R3 = 37%; $\chi^2(2)$ = 0.400, p = 0.818). Moreover, in both hemispheres, electrodes showing poor generalization were clustered near the dorsal portion of the central sulcus, a region corresponding to the arm area of motor cortex. Electrodes showing strong generalization (mostly limited to the left hemisphere) tended to be in dorsal and ventral premotor cortices (PMd and PMv), along with a few in superior parietal cortex. This pattern was also observed when we analyzed all electrodes, rather than restrict the analysis to those showing extreme values. Here we used a continuous measure, correlating the amount of across-arm generalization with the distance (absolute value) from the dorsal aspect of the central sulcus. The correlation was significant in the left hemisphere [$r_{left}$ = 0.46, $p_{left}$ < 0.001] but did not reach significance in the right hemisphere, although the trend was in the same direction [$r_{right}$ = 0.22, $p_{right}$ = 0.068; *Figure 5—figure supplement 1*].

## Target modulation and tuning similarity across arms

To examine the extent of target modulation for the contralateral and ipsilateral arms, we calculated the modulation depth (MD) of each electrode during the instruction and movement phases. The modulation index reflects the amount of variability in the signal captured by target tuning (or target specificity): A modulation index of 0.1 means 10% of the variance is captured by the difference between the response to the four target locations. The modulation values overall were relatively low (*Figure 5D*). However, it should be noted that the reaches were all within the frontoparallel plane which comprise a considerably smaller range of movement compared to studies that use a center-out reaching task. For both electrode types (showing good or poor across-arm generalization), there was a main effect of arm, with ipsilateral modulation lower than contralateral modulation [permutation test: $p_{Generalize\_well}$ < 0.001; $p_{Generalize\_poorly}$ < 0.005]. Both subgroups of electrodes also displayed a main effect of task phase, with the depth of modulation greater during the movement phase compared to the instruction phase [permutation test: $p_{Generalize\_well}$ < 0.001; $p_{Generalize\_poorly}$ < 0.005]. No significant interactions were found for either group.

We also examined the representational overlap between the two arms in terms of their tuning profiles. We computed a tuning similarity index (SI), defined as the sum of squared errors (SSEs) for

average HFA predictions to the same target between the contralateral and ipsilateral arms. An SI of 1 would correspond to identical tuning preferences for the arms whereas an SI of 0 would indicate completely disparate tuning preferences. The similarity data were analyzed with a mixed design permutation test, including the factors task phase and electrode type (good vs. poor generalizers). Electrodes that generalize well across the two arms (predominately found in the left hemisphere) showed more overlap of tuning preferences compared to electrodes that generalized poorly [main effect of generalizability: p < 0.001]. While there was no effect of phase [p > 0.70], the interaction was significant [p < 0.005], with electrode types showing more comparable tuning similarity during instruction and tuning similarity diverging during movement. Simple effects analysis revealed that for electrodes that generalize poorly, tuning similarity was higher during the instruction phase compared to the movement phase [permutation test: p < 0.001]. In contrast, for electrodes that generalize well, tuning similarity was higher during movement compared to instruction [permutation test: p < 0.001]. These analyses demonstrate that a number of electrodes in the left hemisphere encode kinematic variables for both arms, including similar tuning preferences across the two arms, which was especially pronounced during the movement phase.

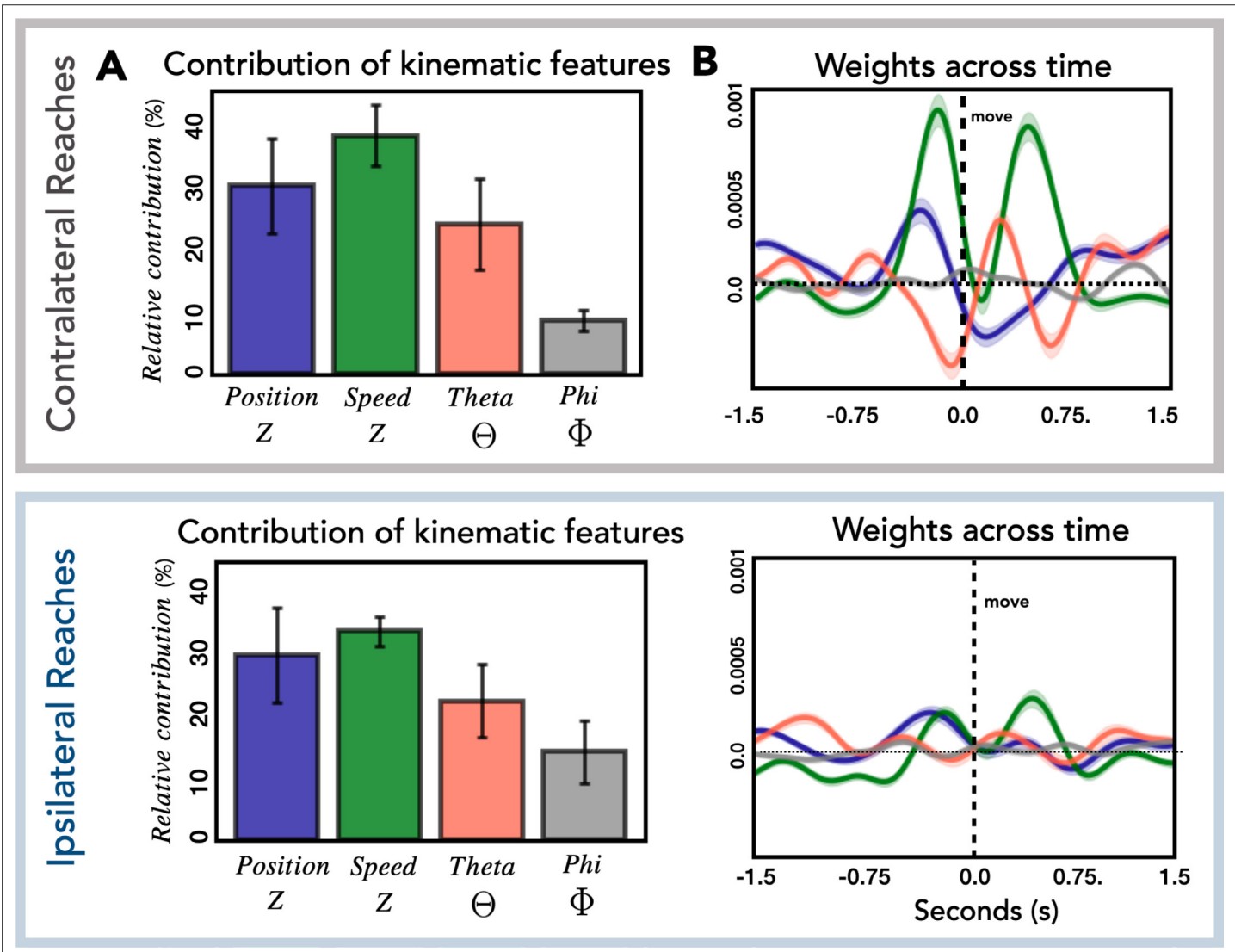

**Figure 6.** Relationship of kinematic features to high-frequency activity (HFA). (**A**) Relative contribution of kinematic features to the encoding model. Average contribution of the four kinematic features for contralateral (top) and ipsilateral (bottom) reaches. Error bars represent the standard deviation across patients. (**B**) Temporal pattern of weights across time. Average weights for all predictive electrodes are plotted for each kinematic feature for contralateral (top) and ipsilateral (bottom) reaches. Error bars represent the standard error across electrodes.

## HFA relationship to kinematics

In the final analysis, we examined how the kinematic features of the movements contribute to the encoding model used to predict HFA. Each of the four kinematic features includes 400 time lags and thus 400 weights that contribute to the model. To obtain a metric of the relative contribution of the features, we calculated the total contribution of each feature and normalized these values by dividing by the total contribution of the four features. The calculation was done for each patient separately and then averaged, with error bars representing the standard deviation across patients (*Figure 6A*). The relative contribution of the four kinematic features was similar for contralateral and ipsilateral reaches. We next examined the temporal profile of the weights (*Figure 6B*) and found that this was also similar for the two conditions, although the average weights for ipsilateral reaches are substantially lower, consistent with the observation of lower performance metrics for ipsilateral reaches across all predictive electrodes (*Figure 2—figure supplement 2*).

As can be seen in *Figure 6A*, speed and position, kinematic features which are associated with timing and movement initiation make a strong contribution to the encoding model (relative contribution: contra = 68%, ipsi = 63%). This is in contrast with the smaller contribution of theta and phi, features which provide information about movement direction (relative contribution: contra = 32%, ipsi = 37%). This result is similar to that observed in single-unit and population activity recorded in premotor and motor cortex of nonhuman primates. *Kaufman et al., 2016* observed that the largest response component was associated with movement timing/initiation rather than features such as movement direction. Similarly, this direction-independent signal occurs twice during sequential movements (*Zimnik and Churchland, 2021*); in our data, speed has two prominent peaks, one occurring before the reach and the second occurring before the return movement. We were surprised to see the markedly differential weighting for the vertical (theta) and horizontal (phi) directional features. We assume this is likely idiosyncratic to the layout of our targets.

We also examined the correspondence between HFA and the kinematic features as a function of whether electrodes generalize well or poorly (*Figure 7*). For electrodes that generalize well, position most closely corresponds to HFA for both contralateral and ipsilateral movements. The maximum cross-correlation for contralateral and ipsilateral movements was found at a lag of 200 and 150 ms, respectively, with HFA leading hand position. For electrodes that generalize poorly, the kinematic feature that most closely corresponds to HFA for both contralateral and ipsilateral movements is speed. For these electrodes, the maximum cross-correlation for contralateral and ipsilateral movements were both at a lag of 200 ms, with HFA again leading the kinematic feature. Although the ipsilateral HFA signals are considerably lower in amplitude, the pattern between HFA and speed is quite similar for both ipsilateral and contralateral movements. The fact that the neural activity from electrodes that generalize poorly (primarily located over M1) correlates well with speed provides additional evidence that a strong component of the HFA ECoG signal is related to timing and movement initiation (*Kaufman et al., 2016*).

## Discussion

Although the most prominent feature of cortical motor pathways is their contralateral organization, unimanual movements are well represented in the ipsilateral hemisphere. Single-unit activity and local field potentials obtained from motor cortex in nonhuman primates (*Ganguly et al., 2009*; *Ames and Churchland, 2019*), as well as ECoG activity in humans (*Bundy et al., 2018*; *Ganguly et al., 2009*; *Wisneski et al., 2008*) can be decoded to predict complex kinematic variables and EMG activity during arm movements of the ipsilateral arm.

Here, we report the results from ECoG data obtained from six patients, three with left hemisphere implants and three with right hemisphere implants, with each patient having at least 17 predictive electrodes. To examine how kinematic features are represented in each hemisphere, we built an electrode-wise encoding model. Such models allow prediction of the full time series for each electrode thus retaining the high spatial and temporal resolution of the intracranial signal. From these metrics we could compare kinematic encoding and across-arm generalization between the two hemispheres as well as the spatial distribution of the information-carrying electrodes within each hemisphere.

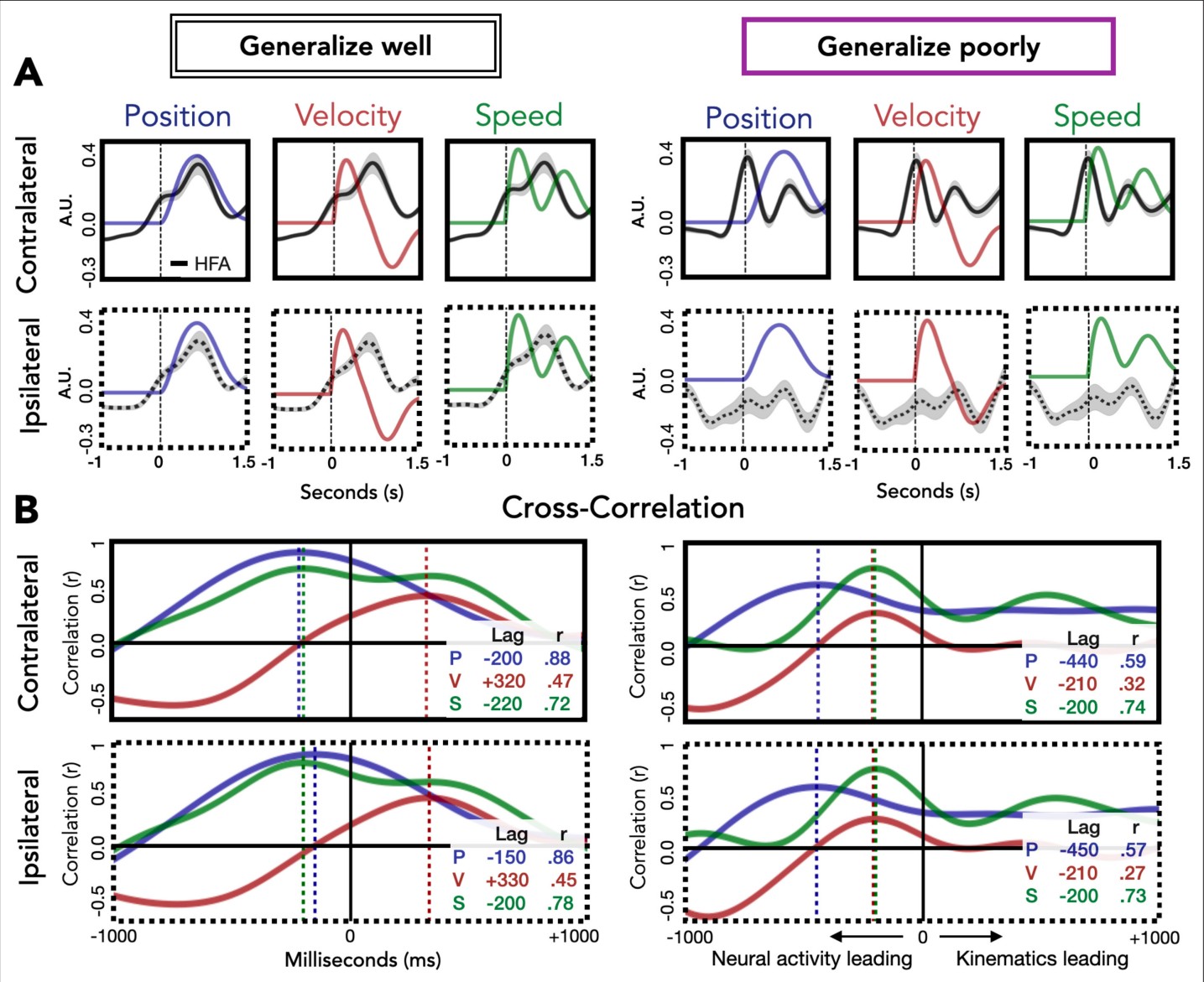

**Figure 7.** Relationship of kinematic features to high-frequency activity (HFA) for electrodes that generalize well or poorly. (**A**) HFA overlaid with kinematics. Each plot shows average HFA activity along with average values for position, velocity, or speed, time-locked to movement onset. (**B**) Cross-correlation between HFA and kinematic features. The cross-correlation between each kinematic feature and the HFA is plotted for electrodes that generalize well or poorly. For each plot, the lag (in ms) yielding the highest correlation between HFA and kinematics is represented by a dashed line with the lag time and correlation coefficient displayed in the inset at the bottom right of each plot.

### Hemispheric asymmetry in movement encoding

We observed a striking asymmetry between the two hemispheres for ipsilateral movement encoding. While contralateral movements were encoded similarly across the two hemispheres, ipsilateral encoding was much stronger in the left hemisphere, an effect that was especially pronounced during movement execution. In addition, there was greater overlap between the representation of contralateral and ipsilateral movement in the left hemisphere compared to the right hemisphere.

The bilateral encoding effect size is quite substantial (Cohen's $d$ = 1.34), exceeding the conventional criterion for a large effect ($d$ = 0.80). Given the size of the hemispheric asymmetry effect, it is surprising that this asymmetry has not been described in previous reports. This may in part reflect the smaller sample size in ECoG studies. For example, in **Bundy et al., 2018**, three of the four patients had left hemisphere grids, leaving a hemisphere analysis dependent on the data from a single right hemisphere patient. Studies with nonhuman primates tend to ignore hemispheric differences, perhaps

because these animals do not show consistent patterns of hand dominance across individuals. One exception here is a study by *Cisek et al., 2003* who reported no hemispheric differences in neural recordings obtained from M1 and PMd during ipsilateral and contralateral arm reaches.

In addition to examining hemispheric differences in the encoding of unimanual movement, we also asked if kinematic features were encoded differently for contralateral and ipsilateral movements by testing across-arm generalization. We categorized electrodes as showing either good across-arm generalization (decrease of up to 20% relative to within-arm performance) or poor across-arm generalization (decrease of more than 50%). This categorization scheme revealed a striking anatomical division, with electrodes showing good across-arm generalization clustering in the left premotor and superior parietal regions and electrodes that generalized poorly clustering in left and right M1. This result does not appear to reflect a sampling bias. There are more left hemisphere electrodes compared to right hemisphere electrodes because two of the left hemisphere patients had high density grid implants. However, the electrodes in our left and right hemisphere samples were similarly distributed over premotor, sensorimotor, and parietal regions (see *Figure 2—figure supplement 1*).

Examining the temporal profile of electrodes that generalize poorly, we found that the divergence between the two arms occurs during instruction, with the ipsilateral trace becoming inhibited relative to the contralateral trace. TMS studies have shown reduced corticospinal excitability of the nonselected hand during movement preparation (e.g., *Leocani et al., 2000*; *Liepert et al., 2001*), an effect that has been hypothesized to reflect inhibition from prefrontal regions to facilitate response selection (*Duque and Ivry, 2009*).

We further examined the spatial tuning of the electrodes. Target tuning in the HFA band was found for both contralateral and ipsilateral movements, although ipsilateral tuning was significantly shallower. Interestingly, electrodes that generalized well across arms had similar spatial/target tuning for each arm. This suggests that for these electrodes, ipsilateral signals are not just encoding generic movement, but encoding movement direction in a similar manner to contralateral signals. A similar overlap in tuning has been observed in single-unit recordings from PMd (*Cisek et al., 2003*) and can be inferred from the across-arm generalization decoding results reported by *Bundy et al., 2018*. In contrast, electrodes that failed to generalize, located primarily in M1 in either left or right hemisphere, exhibited disparate tuning for contralateral and ipsilateral reaches.

## Functional implications of hemispheric asymmetries in movement encoding

By using a delayed response task, we were able to segregate activity into an instruction phase during which the participant was presented with the target location for the forthcoming movement and a movement phase, defined at the onset of the reach. With this design, we found that the encoding model could predict neural activity during the instruction phase based on the kinematics of the forthcoming reach, evidence that the participants were indeed planning the upcoming movement.

This task phase analysis also revealed robust asymmetries between the two hemispheres. There was a main effect of hemisphere, with the left hemisphere displaying stronger bilateral encoding overall compared to the right hemisphere. However, there was also an interaction: In the left hemisphere bilateral encoding was stronger during the movement phase whereas in the right hemisphere bilateral encoding was stronger during the instruction phase. Surprisingly, in the left hemisphere the contralateral bias completely disappeared during the movement phase, with both the contralateral and ipsilateral arms being encoded to the same extent. Stronger bilateral encoding during movement (compared to instruction) is surprising given the spatial distribution of electrodes that encode ipsilateral movement was primarily outside of M1, regions typically associated more with planning than execution (e.g., premotor cortices and parietal cortex).

The asymmetry observed here is in accord with the long-standing recognition of hemispheric asymmetries in praxis. Starting with the classic observations of Liepmann at the turn of the 20th century on the association of the left hemisphere and apraxia (*Liepmann, 1908*, cited in *De Renzi and Lucchelli, 1988*; see also *Schaefer et al., 2007*) and continuing with functional imaging studies in neurotypical populations, a large body of evidence points to a dominant role for the left hemisphere in skilled movement (*Corballis et al., 2012*; *Przybylski and Króliczak, 2017*). This asymmetry is most pronounced in tasks involving functional object use (*Buxbaum et al., 2006*), symbolic gestures (*Xu et al., 2009*), and intransitive pantomimes (*Bohlhalter et al., 2009*). While the neuropsychological and neuroimaging

work have highlighted the involvement of left premotor and parietal cortex in praxis, corresponding asymmetries have also been noted in subcortical structures such as the basal ganglia, cerebellum, and thalamus (*Swinnen et al., 2010*). We note that for our patient population, we are limited to regions of the brain where we have sufficient electrode coverage.

Apraxia, following left hemisphere damage can be manifest in movements produced with either limb (*De Renzi and Lucchelli, 1988*), and are usually associated with lesions that encompass premotor and parietal cortices (*Haaland et al., 2000*). While this asymmetry may be linked to hand dominance (*Ochipa et al., 1989*), functional imagining studies with relatively large sample sizes have shown that handedness only influences the strength of the left hemisphere bias for skilled movement but does not produce a reversal in left handers (*Vingerhoets et al., 2012*; *Verstynen et al., 2005*; *Chettouf et al., 2020*; *Vingerhoets et al., 2013*). Of the six patients tested in the current study, five are right handed and the remaining patient reported being ambidextrous with a slight preference for using the left hand. We note that the results from this patient (L3) did not qualitatively differ from the other two left hemisphere patients.

Ipsilateral encoding was most prominent in the premotor and parietal cortex of the left hemisphere, overlapping with the neural regions implicated in praxis. However, two features of our results do not map on readily to an interpretation that focuses on hemispheric asymmetries in praxis. First, our task involved simple reaching movements, whereas praxis generally encompasses more complex learned movements associated with tool use or symbolic gestures. Second, ipsilateral encoding became more pronounced during movement execution.

Assuming that ipsilateral encoding is indicative of bilateral motor representations, one might have expected, a priori, that hemispheric asymmetries in ipsilateral encoding would be more prevalent during movement planning. We recognize that, even with delayed response tasks, it is overly simplistic to assume that the activity cleanly separates into planning and execution phases. This is especially true with sequential movements where planning effects are observed both prior to and during movement execution (*Ariani and Diedrichsen, 2019*; *Zimnik and Churchland, 2021*). While the experimental task was to reach to a cued target, the participants made out-and-back movements, returning to the home position in a relatively smooth manner (see *Figure 1—figure supplement 1*). It is reasonable to assume that some component of the activity during the primary outward movement was related to planning the return movement. As such, we are hesitant to draw strong inferences about the differences between ipsilateral encoding during action planning and movement execution.

To this point, we have focused on how ipsilateral activity may be reflective of control processes associated with movements of the ipsilateral arm. However, it is possible that this activity is related to postural stabilization of the body during reaching. Indeed, extrapyramidal pathways such as the reticulospinal track have a prominent ipsilateral projection associated with postural control (*Cleland and Madhavan, 2021*), and these pathways receive cortical input. We aimed to reduce postural demands in our task by having the participants seated in an upright hospital bed with the back fully supported; nonetheless, there are surely postural shifts associated with the reaching movement. The clinical setting precluded the use of EMG or video, measurements that would allow a quantitative assessment of the relationship of ipsilateral activity to postural adjustments. We do note a few points that are at odds with a posture-based account of ipsilateral activity. First, we are unaware of evidence suggesting a left hemisphere specialization in postural control. In fact, the evidence suggests that postural instability is more frequently associated with right hemisphere lesions (*Bohannon et al., 1986*; *Spinazzola et al., 2003*). Second, postural adjustments typically precede movement onset (*Belenkii et al., 1967*, cited in *Guiard, 1987*), whereas ipsilateral encoding in our data increased with movement onset. Third, and perhaps most convincing, the encoding model shows that the ipsilateral signals predict reaching kinematics with relatively high precision.

The asymmetry of ipsilateral encoding may be reflective of a prominent role of the left hemisphere in bimanual coordination (*Jäncke et al., 2000*; *Toyokura et al., 1999*; *Maki et al., 2008*; *Serrien et al., 2003*; *Fujiyama et al., 2016*). For example, *Schaffer et al., 2020* observed greater impairments in bimanual coordination following left hemisphere stroke compared to right hemisphere stroke. Interestingly, the impairment was manifest prior to peak velocity, a finding interpreted as a disruption in predictive control. It may be that the left hemisphere makes an asymmetric contribution to interlimb coordination by tracking or predicting where both limbs are in space. As such, the encoding of ipsilateral arm movement might be a form of state representation, a means to keep

track of the state of the ipsilateral arm given that many actions require the coordinated activity of the two limbs. This hypothesis, derived from the current data, is consistent with the increased ipsilateral encoding during the movement phase. The need to monitor the state of the other limb should hold for unimanual gestures performed with either limb.

An important question for future work is to examine how ipsilateral representations in the left hemisphere are affected during more complex movements, including those that involve both limbs. Using fMRI, *Diedrichsen et al., 2013* compared ipsilateral movement representations during unimanual and bimanual movements. Within the primary motor cortex, ipsilateral representations could only be discerned during unimanual movement. However, caudal premotor and anterior parietal regions retained similar ipsilateral representation during uni- and bimanual movement. If the left hemisphere tracks both limbs to facilitate bimanual coordination, we would predict that ipsilateral representations in premotor cortex are retained more strongly in the left hemisphere compared to the right hemisphere when both arms are engaged in the task.

## Conclusion

Using a kinematic encoding model, we observed a striking hemispheric asymmetry, with the left hemisphere more strongly encoding the ipsilateral arm than the right hemisphere, a finding that was apparent during preparation and amplified during movement. This asymmetry was primarily driven by electrodes positioned over premotor and parietal cortices, with strong contralateral encoding for electrodes positioned over sensorimotor cortex. We propose that these networks monitor the state of each arm, a prerequisite for most skilled actions.

# Materials and methods
## Participants

Intracranial recordings were obtained from six patients (two female; five right handed; *Table 1*) implanted with subdural grids as part of their treatment for intractable epilepsy. Data were recorded at three hospitals: University of California, Irvine (UCI) Medical Center (*n* = 2), University of California, San Francisco (UCSF) Medical Center (*n* = 2), and California Pacific Medical Center (CPMC), San Francisco (*n* = 2). Electrode placement was solely determined based on clinical considerations and all procedures were approved by the institutional review boards at the hospitals, as well as the University of California, Berkeley. All patients provided informed consent prior to participating in the study.

## Behavioral task

Participant performed an instructed-delay reaching task while sitting upright in their hospital bed. The participant rested their arms on a horizontal platform (71 cm × 20 cm) that was placed over a standard hospital overbed table. The platform contained two custom-made buttons, each connected to a microswitch. At the far end of the platform (13 cm from the buttons, approximately 55 cm from the participant's eyes), a touchscreen monitor was attached, oriented vertically. Visual targets could appear at one of six locations, four for each arm (*Figure 1A*). The two central locations were used as targets for reaches with either arm; the two eccentric targets varied depending on the arm used. Stimulus presentation was controlled with Matlab 2016a. A photodiode sensor was placed on the monitor to precisely track target presentation times. The analog signals from the photodiode and the two microswitches were fed into the ECoG recording system and were digitized into the same data file as the ECoG data with identical sampling frequency.

Testing of the contralateral and ipsilateral arms (relative to the ECoG electrodes) was conducted in separate experimental blocks that were counterbalanced. To start each trial, the participant placed their left and right index fingers on two custom buttons to depress the microswitches (this indicated they were in the correct position and ready to start the trial). If both microswitches remained depressed for 500 ms, a fixation stimulus was presented in the middle of the screen for 750 ms, followed by the target, a circle (1.25 cm diameter) which appeared in one of the four locations. Another hold period of 900 ms followed in which the participant was instructed to prepare the required movement while the target remained on the screen. If the microswitch was actuated during this hold period, an error message appeared on the screen and the program would advance to the next trial. If the start position was maintained, a compound imperative stimulus was presented at the end of the hold period. This

consisted of an auditory tone and an increase in the size of the target (2.5 cm diameter). The participant was instructed that this was the signal to initiate and complete a continuous out-and-back movement, attempting to touch the screen at the target location before returning back to the platform. The target disappeared when the touchscreen was contacted. The imperative was withheld on 5% of the trials ('catch' trials) to ensure that the participant only responded after the onset of the imperative.

Once back at the home position, the screen displayed the word 'HIT' or 'MISS' for 750 ms to indicate if the touch had occurred within the target zone. The target zone included the 2.5 diameter circle as well as a 1 cm buffer around the target. After the feedback interval, the screen was blank for 250 ms before the reappearance of the fixation stimulus, signaling the start of the next trial. The participants were informed to release either of the buttons at any time they wished to take a break.

Each block consisted of 40 trials (10/target), all performed with a single limb. Blocks alternated between contralateral and ipsilateral arms (relative to the ECoG electrodes), with the order counterbalanced across participants. Each block took approximately 5–6 min to complete. All participants completed at least two blocks with each per arm (*Table 1*).

## Movement analysis and trajectory reconstruction

We used two methods to analyze the movements. For the first method, we recorded key events defined by the release of the microswitch at the start position, time and location of contact with the touchscreen, and return time to the home position, defined by the time at which they depressed the home position microswitch. For the second method, we used the Leap Motion 3D movement analysis system (*Weichert et al., 2013*) to record continuous hand position and the full movement trajectory (sampling rate = 60 Hz). Although the Leap system is a lightweight video-based tracking device that is highly mobile, the unpredictable environment of the ICU led to erratic recordings from the Leap system. For example, patients frequently had intravenous lines in one or both hands which obstructed the visibility of the hand and interfered with the ability of the Leap system to track the hand using their built-in hand model. This resulted in lost samples and therefore satisfactory kinematic data were obtained from only a subset of conditions collected from patients using the Leap system.

Given the limitations with the Leap data, we opted to use a simple algorithm to reconstruct the time-resolved hand trajectory in each trial, estimating it from the event-based data obtained with the first method. We used a beta distribution to estimate the velocity profile of the forward and return reach based on reach times and the travel distance (sampling rate = 100 Hz). We opted to use a beta distribution because this best matched the velocity profiles of the data obtained with the Leap system.

For conditions that had clean kinematic traces (no lost samples) from the Leap system, we compared the estimated kinematic profiles with those obtained with the Leap system. There was a high correlation between the two datasets ($r = 0.98$ for position in the Z dimension; $r = 0.93$ for velocity in the Z dimension; *Figure 1—figure supplement 1*). We note that our method of estimating the trajectories results in a smoothed version of the movement, one lacking any secondary or corrective movements that are sometimes observed when reaching to a visual target (*Suway and Schwartz, 2019*). We believe this is still a reasonable estimation given the high correlation with the continuous Leap data, and the fact that participants had ample time to prepare the movements and were instructed and observed to make ballistic movements by the experimenter who was present for all recording sessions (CMM).

## Electrode localization

Grid and strip electrode spacing was 1 cm in four patients and 4 mm in the two other patients. The electrode locations were visualized on a three-dimensional reconstruction of the patient's cortical surface using a custom script that takes the postoperative computed tomography scan and coregisters it to the preoperative structural magnetic resonance scan (*Stolk et al., 2018*).

## Neural data acquisition and preprocessing

Intracranial EEG data and peripheral data (photodiode and microswitch traces) were acquired using a Nihon Kohden recording system at UCI (128 channel, 5000 Hz digitization) and CPMC (128 channel, 1000 Hz digitization rate), and two Tucker Davis Technologies recording systems at UCSF (128 channel, 3052 Hz digitization rate).

Offline preprocessing included the following steps. First, if the patient's data were not sampled at 1000 Hz (UCI and UCSF recording sites), the signal from each electrode was low-pass filtered at 500 Hz using a Butterworth filter as an antialiasing measure before downsampling to 1000 Hz. Electrodes were referenced using a common average reference. Each electrode was notch filtered at 60, 120, and 180 Hz to remove line noise. The signals were then visually inspected and electrodes with sustained excessive noise were excluded from further analyses. The signals were also inspected by a neurologist (RTK) for epileptic activity and other artifacts. Electrodes that had pathological seizure activity were also excluded from the main analyses. Out of 752 electrodes, 82 were removed due to excessive noise and 5 were removed due to epileptic activity, resulting in a final dataset of 665 electrodes. Catch trials and unsuccessful reaches were not included in the analyses.

From the cleaned dataset, we extracted the HFA instantaneous amplitude using a Hilbert transform. To account for the $1/f$ power drop in the spectrum, we divided the broadband signal into five narrower bands that logarithmically increased from 70 to 200 Hz (i.e., 70–86, 86–107, 107–131, 131–162, and 162–200 Hz), and applied a band-pass filter within each of these ranges. We then took the absolute value of the Hilbert transform within each band-pass, performed a z-score transformation, and averaged the five values. z-Scoring was performed after concatenating all the blocks for each patient, ensuring that we did not obscure possible amplitude differences across the two arms. As a final step, the data were downsampled to 100 Hz to reduce computational load (e.g., number of parameters in the encoding model, see below). HFA amplitude fluctuations (envelope; are evident at lower frequencies *Canolty et al., 2006*; *Pei et al., 2011*).

## Feature selection

Four estimated kinematic features were used to predict HFA (*Figure 1B*, left). The first two features were position and speed in the *Z* dimension. This dimension captures variability related to movement that is relatively independent of target location (i.e., along the axis between the participant and touchscreen). The second pair of features were spherical angles that define the specific target locations (*Figure 1A*, right). Features were selected to reduce collinearity and redundancy in the encoding model. Because we include time lags for each kinematic feature, derivatives can emerge from the linear model (e.g., velocity and acceleration can be created from position); thus, velocity and acceleration were not included as additional features. Speed is a nonlinear transformation of position and is added as a separate feature.

## Kinematic encoding model

The estimated kinematic features were used to predict the HFA for each electrode (*Figure 1F*). We created a 4 × 400 feature matrix by generating a time series for each feature by time lagging the values of the selected feature relative to the neural data, with lags extending from 2 s before movement onset to 2 s after movement onset (sampling rate at 100 Hz). This wide range of lags serves two purposes. First, it provides a way to compensate for the anticipated asynchrony between neural data and movement kinematics. Second, it allowed us to evaluate HFA activity during the instructed-delay (beginning ~1.5 s before movement onset) period as well as during movement. HFA at each time point [HFA($t$)] was modeled as a weighted linear combination of the kinematic features at different time lags, resulting in a set of beta weights, $b1, …, b400$ per kinematic feature. To make the beta weights scale-free, the kinematic features and neural HFA were z-scored before being fit by the model.

## Fitting

Regularized (ridge) regression (*Hoerl and Kennard, 1970*) was used to estimate the weights that map each kinematic feature ($X$) to the HFA signal ($y$) for each electrode, with $\lambda$ being the regularization hyperparameter:

$$\hat{\beta} = \left( X^T X + \lambda I \right)^{-1} X^T y$$

For within-arm model fitting, the total dataset consisted of all clean, successful trials performed with either the ipsilateral or contralateral arm (each arm was fit separately). Nested fivefold cross-validation was used to select the regularization hyperparameter on inner test sets (validation sets) and assess prediction performance on separate, outer test sets. At the outer level, the data were

partitioned into five mutually exclusive estimation and test sets. For each test set, the remaining data served as the estimation set. For each outer fold, we further partitioned our estimation set into five mutually exclusive inner folds to train the model (80% of estimation set) and predict neural responses across a range of regularization values on the validation set (20% of estimation set). For each inner fold, the regularization parameter value was selected that produced the best prediction as measured by the linear correlation of the predicted and actual HFA. The average of the selected regularization parameters across the five inner folds was computed and used to calculate the prediction of the HFA on the outer test set. This procedure was done at the outer level five times. Our primary measure is held-out prediction performance ($R^2$), which we quantified as the squared linear correlation between the model prediction and the actual HFA time series, averaged across the five mutually exclusive test sets.

To be considered as predictive, we established a criterion that an electrode must account for at least 5% of the variability in the HFA signal ($R^2 > 0.05$) for either ipsilateral or contralateral reach (**Downey et al., 2020**). Electrodes not meeting this criterion were not included in subsequent analyses.

For across-arm model fitting, the same procedure was used except the test set was partitioned from the total dataset of the other arm. We partitioned the data in this manner (80% estimation, 20% test) to make the fitting procedure for the across-arm model comparable to that employed in the within-arm model.

## Tuning modulation and similarity across arms

MD of target tuning was calculated as the standard deviation of the mean HFA predictions for each of the four target locations:

$$MD = \sum_{i=1}^{n} \frac{(x_i - \bar{x})^2}{n}$$

where $x$ is the average HFA prediction, $x_i$ is the average HFA prediction for each of the four target positions. $i$ iterates through the target locations and $n$ is the total number of target locations. This process was done separately for contralateral and ipsilateral HFA predictions.

To assess similarity in tuning across the two arms, we computed the SSEs for average HFA predictions to the same target between the contralateral and ipsilateral arms. This calculation was computed for each electrode as follows:

$$SSE_e = \sum_{i=1}^{n} (contra_i - ipsi_i)^2$$

where *contra* and *ipsi* are average HFA predictions for a given target location reached with either the contralateral or ipsilateral arm. Note that for this calculation $n$ is limited to the two positions that served as target locations for both arms.

This metric was only calculated for the two central targets, the targets common to both arms (the two eccentric target locations varied depending on the arm used). These values were scaled from 0 to 1 based on the minimum and maximum values of SSE across all electrodes. SSE represents a metric of dissimilarity; to calculate a similarity index (SI), we subtracted the scaled SSE values from 1:

$$SI = 1 - \frac{SSE_e - min(SSE_e)}{max(SSE_e) - min(SSE_e)}$$

Thus, higher SI represents more similar average predictions.

## Separating instruction and movement phases

The encoding model was run to predict the full HFA time course. To compare model prediction performance during different phases of the task, the data were epoched into instruction and movement phases, using event markers recorded in the analog channel (i.e., cue onset and movement onset). Epochs of the same task phase were concatenated together, and prediction performance was operationalized as the square of the Pearson's correlation between the predicted and actual HFA for each task phase.

## Permutation-based linear mixed-effects model

Linear mixed-effects models were carried out in RStudio using software packages lme4 and permlmer (*Bates et al., 2014*; *Lee and Braun, 2012*). Each mixed-effects model used participant as a random effect and experimental variables (e.g., reaching arm, hemisphere) as fixed effects. The models were used to predict performance from the kinematic encoding model for all predictive electrodes. Nested models were created to assess the effect of the fixed factors, with the null model using patient as a random effect to predict encoding values. For the nesting, fixed factors were added to the model to assess if each new factor improved prediction above that obtained with the null model using a permutation-based method (*Anderson and Braak, 2003*, *Lee and Braun, 2012*). Interactions were tested by comparing a model in which the fixed effects were restricted to have additive effects to a model that could have both multiplicative and additive effects.

## Assessing the contribution of kinematic features in the encoding model

Average weights for each patient were calculated for all predictive electrodes for each kinematic feature. Because each feature has 400 time lags, there are 400 weights per feature in the full model. To assess the relative contribution of each feature, we calculated the sum of the weights across time lags after taking the absolute value of each weight (since negative weights are as informative as positive weights). The sum of each feature was then plotted as a proportion against the total sum of all weights (after taking the absolute value) to assess the relative contribution. To capture the temporal profile of the weights, the average was taken for all predictive electrodes for all patents.

## Assessing explainable variance in the encoding model

There are several methods to assess prediction performance for encoding models (*Lage-Castellanos et al., 2019*; *Nunez-Elizalde et al., 2019*; *Schoppe et al., 2016*). A drawback with the Pearson's product-moment correlation coefficient is that it does not distinguish between explainable variability and response variability (*Hsu et al., 2004*). To better capture this distinction, we calculated $CC_{norm}$, a measure which normalizes by signal power (SP) to account for response variability (*Schoppe et al., 2016*):

$$CC_{norm} = \frac{Cov(y, \hat{y})}{\sqrt{Var(\hat{y})}} \sqrt{\frac{1}{SP}}$$

where $y$ is neural activity and $\hat{y}$ are predictions from the model, and SP is:

$$SP = \frac{Var\left(\sum_{n=1}^{N} R_n\right) - \sum_{n=1}^{N} Var(R_n)}{N(N-1)}$$

where $R_n$ is the neural time series for the $n$th trial and $N$ is the total number of trials.

We plot the density estimates of all predictive electrodes for both the Pearson's correlation coefficient ($CC_{abs}$) and $CC_{norm}$ (see *Figure 2—figure supplement 2*). We find similar values for the two metrics, suggesting that the number of trials in our study and SP is sufficient for the encoding model to capture the majority of explainable variance. (Calculations of Pearson's correlation coefficient were always taken from held-out trials.)

## Calculating distance from dorsal central sulcus

For each patient, 30 discrete ($x$, $y$) coordinates were manually demarcated along the central sulcus on individual MRI scans. The 30 points were then interpolated to create a line traversing the central sulcus for each individual. The dorsal aspect of the central sulcus was defined as all points dorsal to the midpoint of the central sulcus. We then calculated the absolute distance between each electrode and the closest point on the dorsal aspect of the central sulcus (our interpolated line).

## Acknowledgements

We want to thank Anwar Nunez-Elizalde for his invaluable guidance building the encoding model, Ian Greenhouse for building the first prototype rig to bring into the ICU and William Liberti for his advice with the figures. This research was supported by the National Institutes of Health: NS097480.

## Additional information

### Competing interests

Richard B Ivry: Senior editor, *eLife*. The other authors declare that no competing interests exist.

### Funding

| Funder | Grant reference number | Author |
|---|---|---|
| National Institutes of Health | NS097480 | Robert Thomas Knight<br>Richard B Ivry<br>Jose Carmena |

The funders had no role in study design, data collection, and interpretation, or the decision to submit the work for publication.

### Author contributions

Christina M Merrick, Conceptualization, Investigation, Methodology, Software, Visualization, Writing - original draft; Tanner C Dixon, Assaf Breska, Conceptualization, Writing - review and editing; Jack Lin, Edward F Chang, David King-Stephens, Kenneth D Laxer, Peter B Weber, Resources; Jose Carmena, Conceptualization, Funding acquisition, Methodology, Supervision; Robert Thomas Knight, Conceptualization, Funding acquisition, Supervision, Writing - review and editing; Richard B Ivry, Conceptualization, Funding acquisition, Methodology, Supervision, Writing - review and editing

### Author ORCIDs

Christina M Merrick http://orcid.org/0000-0003-4966-9138
Tanner C Dixon http://orcid.org/0000-0003-3151-3362
Assaf Breska http://orcid.org/0000-0002-6233-073X
Edward F Chang http://orcid.org/0000-0003-2480-4700
Robert Thomas Knight http://orcid.org/0000-0001-8686-1685
Richard B Ivry http://orcid.org/0000-0003-4728-5130

### Ethics

Intracranial recordings were obtained from six patients implanted with subdural grids as part of their treatment for intractable epilepsy. Data were recorded at three hospitals: University of California, Irvine (UCI) Medical Center, University of California, San Francisco (UCSF) Medical Center, and California Pacific Medical Center (CPMC), San Francisco. Electrode placement was solely determined based on clinical considerations. All procedures were approved by the Institutional Review Board at every site as well as by the Committee for Protection of Human Subjects the University of California, Berkeley (protocol number: 2010-02-783) and conducted in accordance with the Declaration of Helsinki. All patients provided informed consent prior to participating in the study.

### Decision letter and Author response

Decision letter https://doi.org/10.7554/eLife.69977.sa1
Author response https://doi.org/10.7554/eLife.69977.sa2

## Additional files

### Supplementary files

• Transparent reporting form

### Data availability

De-identified ECoG data and task kinematics are available on Zenodo at https://doi.org/10.5281/zenodo.4761390. Code to run these data through the kinematic encoding model can be found at https://github.com/cmerrick15/Asymmetry_ECoG_dataset copy archived at swh:1:rev:25639ec88f6e0279c4925edea6b4927c082815f6.

The following dataset was generated:

| Author(s) | Year | Dataset title | Dataset URL | Database and Identifier |
| --- | --- | --- | --- | --- |
| Merrick C | 2021 | ECoG_Hemispheric_Asymmetry_dataset | https://doi.org/10.5281/zenodo.4761390 | Zenodo, 10.5281/zenodo.4761390 |

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
