## [Editor Report]

Based on a rare set of human intracranial recordings, this paper by Merrick and colleagues asks how the neural processes that generate arm movements are distributed across cortical hemispheres. The authors demonstrate that these contributions are not equal. The key result is that the left hemisphere shows stronger bilateral representations than the right hemisphere. This effect is present during movement preparation but is further accentuated during movement execution. Taken together, the findings provide important new insight into the hemispheric asymmetries that underlie manual actions.

---

## [Decision Letter]

**Decision letter after peer review:**

Thank you for submitting your article "Left Hemisphere Dominance for Bilateral Kinematic Encoding in the Human Brain" for consideration by *eLife*. Your article has been reviewed by 3 peer reviewers, and the evaluation has been overseen by Andrew Pruszynski as the Reviewing Editor and Tirin Moore as the Senior Editor. The following individuals involved in review of your submission have agreed to reveal their identity: Stephan Swinnen (Reviewer #1); Atsushi Yokoi (Reviewer #2).

Essential revisions:

1. Provide more details about the electrode positions and more specificity about the brain areas covered: asymmetries in number of electrodes in each hemisphere and their specific location, variability in electrode coverage, electrode coverage for each hemisphere group, subregions of premotor and parietal cortex being covered (if possible).

2. Importantly, clarify and address the impact of the variability in electrode placement on the statistical tests and ultimately on the conclusions being drawn.

3. Elaborate on the topic of bimanual control. Presently this topic is mentioned in the Discussion section but should be worked out in more detail. Some relevant references in this domain are proposed. In general, several relevant references as proposed in the reviews should be added and considered in more detail.

*Reviewer #1 (Recommendations for the authors):*

1. The authors refer to medical imaging research that has identified evidence for movement representations common to both upper limbs. In this respect, the authors may be interested in complementary fMRI research involving complex coordination skills performed with the left and right body side, pointing to the critical importance of shared activations in the left premotor-parietal complex for movements of the right and left body side. The findings of the latter study are consistent with the findings presented in the ECoG study in spite of the different imaging techniques. Nevertheless, in spite of the convergent findings, this fMRI study also points to additional candidate brain regions for shared movement representations by the left and right hand that cannot be revealed by ECoG (Swinnen et al., 2010).

2. It is a pity that a large amount of movement kinematic data was lost due to problems with the motion registration equipment in interaction with the patients. Nevertheless, the cross-validated kinematic encoding model is very helpful and provides a reasonable representation of these simple reaching movements.

3. Even though unimanual movements are studied, the participants are supposed to keep their finger on the resting switch for the nonmoving hand. Is it possible that this position serves a postural control function to support the movement of the contralateral arm that reaches for the target? If so, could this account for similarities in neural activity for controlling both hands?

4. Is it possible to provide more detailed information about the whole body position of the participants during performance of these reaching tasks. This information is interesting from the perspective of postural control mechanisms (via trunk and contralateral body side), as required for enabling these focal arm reaching movements.

5. In the Table on p. 4, remarkable differences in RT are evident among the 6 participants. Can the authors provide an explanation for the relatively long RTs in some of the patients? Can this still be considered normal behavior?

6. In the figure captions, all the (sub)graphs are explained in meticulous detail allowing the reader to understand the messages that the authors wish to convey. Overall, the manuscript is excellently written and directs the reader to the focal results in relation to the information provided in the figures.

7. On p. 12, it is stated that the poor generalization of some electrodes is generally due to the temporal divergence of the two arms during instruction, where the ipsilateral trace becomes inhibited compared to the contralateral trace. Do the authors refer to an active inhibition mechanism as evidenced from TMS research during movement preparation in which neural excitation is first suppressed and then followed by a phase in which the excitation level of the arm to the moved is increased while that for the nonmoving arm is further depressed prior to movement initiation (Duque and Ivry, 2009).

8. In the Discussion section, the authors reflect on research from bimanual movement control to identify potential accounts for the bilateral encoding architecture. However, the literature to which they refer leaves room for alternative interpretations. Perhaps, the authors may be interested in a neural model of bimanual control based on TMS interhemispheric connectivity research between PMd and M1 in which left PMd (and not right PMd) is hypothesized to be in charge of the division of labor assigned to the left and right hands during the preparation of bimanual movement (Fujiyama et al., 2016a,b). The latter bimanual work is generally consistent with the hemispheric asymmetry evidence provided in the present manuscript but also reflects on the specific function assigned to the left premotor cortex during bimanual control. This suggests potentially alternative reasons for the finding that electrodes that generalized well across arms were primarily located in premotor regions of the left hemisphere.

References:

Duque, J., Ivry, R.B. (2009) Role of corticospinal suppression during motor preparation. Cerebral Cortex, 19, 2013-2024.

Swinnen, S., Vangheluwe, S., Wagemans, J., Coxon, J., Goble, D., Van Impe, A., Sunaert, S., Peeters, R., Wenderoth, N. (2010). Shared neural resources between left and right interlimb coordination skills: The neural substrate of abstract motor representations. NeuroImage, 49 (3), 2570-2580.

Fujiyama, H., Van Soom, J., Rens, G., Gooijers, J., Leunissen, I., Levin, O., Swinnen, S. (2016a). Age-related changes in frontal network structural and functional connectivity in relation to bimanual movement control. Journal of Neuroscience, 36 (6), 1808-1822.

Fujiyama, H., Van Soom, J., Rens, G., Cuypers, K., Heise, K., Levin, O., Swinnen, S. (2016b). Performing two different actions simultaneously: the critical role of interhemispheric interactions during the preparation of bimanual movement. Cortex, 77, 141-154.

*Reviewer #2 (Recommendations for the authors):*

Personally, I feel the term 'bilateral kinematic encoding' (or 'bilateral encoding'), as it appears in the title or text, is slightly ambiguous, as the 'dominance' for 'bilateral kinematic encoding' can imply that both the contra- and ipsilateral encoding were stronger than the other hemisphere. However, the contralateral kinematic encoding was not explicitly tested in the current paper. It is entirely up to the authors, but I think 'ipsilateral kinematic encoding' would be less confusing than 'bilateral kinematic encoding.' The authors could otherwise compare the contralateral kinematic encoding between the hemispheres.

It was not clear from the text whether the electrodes were pooled over the three patients for each hemisphere group or the patients were still treated as a random effect. Please clarify this point. In either case, it would be informative to show the breakdown of these electrodes in terms of patients (e.g., page 5, line 25 "665" electrodes, page 7, line 11, "141", "75"). A summary table of the electrodes indicating the participants and regions they belong to would be very helpful for readers. As the co-registration process of electrodes on the reconstructed cortical surface indicated by Stolk et al. (2018) uses the freesurfer, it would be relatively easy to find in which of the freesurfer-defined regions, such as Destrieux Atlas (Destrieux et al., Neuroimage, 2010), each of the electrodes covered. Please consider.

Please consider using different colors or marker types for different patients for the scatter plots displayed in Figures 2-5, as this is more informative to readers.

A recent line of studies (Ariani and Diedrichsen, J Neurophysiol, 2019; Ariani et al., J Neurophysiol, 2020) emphasizes the significant contribution of online planning in the production of sequential movement. In this context, the even stronger bilateral shared encoding observed during the movement phase might partially reflect the online planning of return movements. As this return movement is always to the same starting position (target), higher similarity could be explained in part by this component. Please discuss.

Please specify the reason why the authors excluded the peripheral targets for the cross-arm generalization analysis. These targets could provide further insight into under which coordinate frame (i.e., extrinsic or intrinsic) the contra- and ipsilateral encoding overlapped (e.g., Wiestler et al., J Neurosci, 2014; Haar et al., J Neurosci, 2017).

Although up to the authors, it would be recommended to report "noise ceiling" for each electrode (e.g., Schoppe et al., Front Comput Neurosci, 2016). The noise ceiling would also help interpret whether poor prediction accuracy is due to the low SN ratio or the inappropriate model in the across-arm generalization test.

Figure 5B: It would be more informative also to show all other electrodes (or areas covered by each electrode grid, for instance, by colored rectangles) to indicate the coverage for each hemisphere group.

Similarly, displaying the simple ipsilateral encoding strength (shown in Figure 2) in the same format of Figure 5B (categorized by generalization index) would also be informative for comparing the shared and unshared ipsilateral encoding.

Schaffer et al. (2020) was not found in the reference list (Schaffer and Sainburg, 2021 was listed instead). Please list the proper reference.

Please report the age data of the participants in the Methods.

Please provide more details about the statistical analyses in the Methods.

Page 20, line 1: Please specify what the MD, n, i, and x indicate.

Page 20, line 5: Please specify what the SSEe, n, i, contra_i_, and ipsi_i_, indicate.

Page 20, line 12: "min" is omitted before "(SSEe)" in the denominator.

*Reviewer #3 (Recommendations for the authors):*

1. I believe that the recording sites between hemispheres must be matched; this is the only way to ensure that the inter-hemispheric differences reported are not simply due to the intra-hemispheric differences reported in Supplementary Figures 1-2.

2. I'm a bit confused as to why the landmark used in Supplementary Figures 1-2 is the dorsal aspect of the central sulcus. I might expect that premotor electrodes would generalize better (and exhibit more bilateral encoding) than M1 electrodes, but your analysis seems to make the argument that face M1 electrodes also generalize better than arm M1 electrodes. Could you provide a bit more interpretation of this empirical result?

3. I don't quite get the relevance of Churchland 2012 results to the trace in Figure 5C. Churchland et al. used only outward reaches in the 2012 study, whereas your subjects are performing out-and-back movements. The trace in Figure 5C is more similar to the trigger signal reported in Kaufman, et al. In fact, in monkeys making movement sequences (comparable to your out-and-back movements), Zimnik and Churchland (2021) found a biphasic trigger signal that looks pretty similar to the trace shown in 5C. To that end, it might be useful to plot the across-subject hand speeds in 5C. If the ECoG signal plotted is related to the trigger signal, then it should have a time-course similar to the subjects' hand speed.

4. Related to 5C, wouldn't it make more sense to plot the actual HFA amplitude, rather than the predicted HFA amplitude? I understand that the two will likely be very similar, but 5C is making a point about the true neural activity.

[Editors' note: further revisions were suggested prior to acceptance, as described below.]

Thank you for resubmitting your work entitled "Left Hemisphere Dominance for Bilateral Kinematic Encoding in the Human Brain" for further consideration by *eLife*. Your revised article has been evaluated by Tirin Moore as the Senior Editor, a Reviewing Editor, and the original reviewers.

The manuscript has been improved but there are some remaining issues that need to be addressed, as outlined below:

1. All reviewers were very happy with the revisions and think the paper is strong. Please see the remaining comments of Reviewer #2 especially with respect to statistical support for some of the claims.

*Reviewer #1 (Recommendations for the authors):*

The authors have made a profound and detailed attempt to comply with the questions raised by the reviewers. Accordingly, as far as I am concerned, the paper is acceptable for publication in *eLife*.

This is important work in which data is obtained from a unique set of patients to reveal the control of ipsilateral and contralateral arm movement in a very detailed manner. It complements previous work collected from nonhuman primates.

*Reviewer #2 (Recommendations for the authors):*

The authors adequately addressed most of my previous concerns. I have one further recommendation and a comment related to the modification.

The authors need some statistical support for their claims regarding (i) hemispheric equality in regional distribution of electrodes (lines 226-228) and (ii) hemispheric difference in the number of electrodes with good/poor across-arm generalization (lines 412-414, 439-441, and 590-597).

(i) Hemispheric equality in regional distribution of electrodes (lines 226-228):

"When the categorization data were pooled across the three left hemisphere and three right hemisphere patients, the proportion of electrodes was similar in premotor, sensorimotor motor, and parietal regions." (Lines 226-228)

This can be tested by using, for example, multinomial goodness-of-fit tests using expected frequency for each region estimated from the data from either hemisphere. Reporting the number or proportion (%) for these three regions in the main text, in addition to the bar graphs, would also be helpful.

(ii) Hemispheric difference in number of electrodes with good/poor across-arm generalization (lines 412-414, 439-441, and 590-597):

"This categorization scheme revealed a striking anatomical division, with electrodes showing good across-arm generalization clustering in the left premotor and superior parietal regions and electrodes that generalized poorly clustering in left and right M1. This result does not appear to reflect a sampling bias. There are more left hemisphere electrodes compared to right hemisphere electrodes because two of the left hemisphere patients had high density grid implants. However, the electrodes in our left and right hemisphere samples were similarly distributed over premotor, sensorimotor and parietal regions (see Supplementary Figure 2)." (Lines 590-597)

The fact that the electrodes in the left and right hemispheres have similar distribution over these regions does not automatically protect from the risk of finding more electrodes in the left hemisphere due to the hemispheric bias in electrode number, and need formal statistical tests. Similarly to the first point above, this can be tested by using goodness-of-fit tests (e.g., chi-squared test, likelihood-ratio G test, etc.) using the expected number of electrodes that should be found in the same region in the other hemisphere under the null-hypothesis of hemispheric symmetry (e.g., if 20 electrodes are found in the left hemisphere, 74*20/141 electrodes should also be found in the right).

*Reviewer #3 (Recommendations for the authors):*

Merrick et al. were admirably responsive to all reviewer comments, and the manuscript has been greatly improved as a result. I have no further critiques.

---

## [Author Response]

Essential revisions:1. Provide more details about the electrode positions and more specificity about the brain areas covered: asymmetries in number of electrodes in each hemisphere and their specific location, variability in electrode coverage, electrode coverage for each hemisphere group, subregions of premotor and parietal cortex being covered (if possible).

We categorized the electrode positions for all six patients using the Desikan-Killiany atlas and calculated the proportion of electrodes that were centered over subregions within premotor, sensorimotor and parietal regions. While there are more electrodes in left hemisphere compared to the right hemisphere (left 141; right 75), the proportions of electrodes in these areas is comparable. We have added Supplementary Figure 2. This analysis is also addressed in the Results section (page 8, lines: 200-227).

2. Importantly, clarify and address the impact of the variability in electrode placement on the statistical tests and ultimately on the conclusions being drawn.

Given that there is comparable regional electrode coverage in the left and right hemisphere across patients, it is unlikely that the hemispheric differences are confounded by regional differences; as such, we do not think it essential to include subregion as a factor in our model. We also note that not all of the patients have electrodes over each subregion and thus, adding subregion as a fixed factor would reduce power. For these reasons, we focus on continuous measures such as the distance from the central sulcus rather than categorial measures, allowing us to include all task-relevant electrodes in these analyses. We also think this approach is more precise because the values are based on individual MR scans instead of parcellations that would be derived with respect to a reference MNI brain and then fit to each individual scan.

As suggested by Reviewer 2, we agree that it is important to control for patient variability in our statistical tests. We now use a permutation-based mixed effects model which has fixed effects comprising our experimental factors (hemisphere, reaching arm, task-phase, generalization – depending on the specific statistical test) and a random effect of patient. This allows us to address the nested structure of our dataset (electrodes are nested within patient) and provides a way to consider the effect of patient variability. Including patient as a random effect did not change any of the results. We have revised the Results section to report these tests. We have also updated Figures 2-4, using color to indicate the electrodes for each patient, allowing the reader to readily see the variation across patients. We appreciate the prod from Reviewer # 2, with the changes adding more transparency about variability across patients, both in terms of the statistical tests and figures.

3. Elaborate on the topic of bimanual control. Presently this topic is mentioned in the Discussion section but should be worked out in more detail. Some relevant references in this domain are proposed. In general, several relevant references as proposed in the reviews should be added and considered in more detail.

Based on feedback from all three reviewers we have updated the Discussion section on the issue of bimanual control. We detail these changes in the responses below to the comments of the reviewers.

Reviewer #1 (Recommendations for the authors):1. The authors refer to medical imaging research that has identified evidence for movement representations common to both upper limbs. In this respect, the authors may be interested in complementary fMRI research involving complex coordination skills performed with the left and right body side, pointing to the critical importance of shared activations in the left premotor-parietal complex for movements of the right and left body side. The findings of the latter study are consistent with the findings presented in the ECoG study in spite of the different imaging techniques. Nevertheless, in spite of the convergent findings, this fMRI study also points to additional candidate brain regions for shared movement representations by the left and right hand that cannot be revealed by ECoG (Swinnen et al., 2010).

This is a great point given that ECoG provides a restricted view of the brain, one that is clinically determined. We believe that our addition to the text addresses this issue (page 19, lines: 647-652).

2. It is a pity that a large amount of movement kinematic data was lost due to problems with the motion registration equipment in interaction with the patients. Nevertheless, the cross-validated kinematic encoding model is very helpful and provides a reasonable representation of these simple reaching movements.3. Even though unimanual movements are studied, the participants are supposed to keep their finger on the resting switch for the non-moving hand. Is it possible that this position serves a postural control function to support the movement of the contralateral arm that reaches for the target? If so, could this account for similarities in neural activity for controlling both hands?

We have added a paragraph to the Discussion to address the possibility that some aspect of ipsilateral encoding may be related to postural stabilization (page 20-21, lines: 685-703).

4. Is it possible to provide more detailed information about the whole body position of the participants during performance of these reaching tasks. This information is interesting from the perspective of postural control mechanisms (via trunk and contralateral body side), as required for enabling these focal arm reaching movements.

The participants were tested in their hospital bed, with the bed positioned so that the patient was in upright, “seated” position (legs extended). We did not record EMG from potential postural muscles and thus cannot examine the postural question in a quantitative manner. As such, we have opted to note this limitation in the Discussion where we discuss the postural hypothesis (page 20, lines: 693-695). This would be a good issue to tackle directly in future studies.

5. In the Table on p. 4, remarkable differences in RT are evident among the 6 participants. Can the authors provide an explanation for the relatively long RTs in some of the patients? Can this still be considered normal behavior?

The Reviewer is correct in noting the large between-participant variability in RT. This is likely a consequence of the task instructions: We did not emphasize speed, letting the patients work at their own pace and putting the emphasis on accuracy. The only constraint was that the message “Too Slow” appeared if the participant did not initiate the movement within 2.5 s. This was really there in case the patient’s attention lapsed (and rarely happened). Our experience is that RT demands should be minimized with ECoG patients given their clinical state. We also had our plan to fit the kinematics on an individual basis and did not anticipate wanting to average signals across patients. For this reason, we did not see RT variability as a major concern. We have modified the text in the results to note that participants were instructed to move at their own pace and focus on accuracy (page 4, lines: 141-143).

6. In the figure captions, all the (sub)graphs are explained in meticulous detail allowing the reader to understand the messages that the authors wish to convey. Overall, the manuscript is excellently written and directs the reader to the focal results in relation to the information provided in the figures.7. On p. 12, it is stated that the poor generalization of some electrodes is generally due to the temporal divergence of the two arms during instruction, where the ipsilateral trace becomes inhibited compared to the contralateral trace. Do the authors refer to an active inhibition mechanism as evidenced from TMS research during movement preparation in which neural excitation is first suppressed and then followed by a phase in which the excitation level of the arm to the moved is increased while that for the non-moving arm is further depressed prior to movement initiation (Duque and Ivry, 2009).

This is a very interesting point and something that we had discussed but opted to not include in the original submission. Inspired by the comment, we have now added text to the Discussion to discuss this possible mechanism of inhibition (page 18, lines 598-604).

8. In the Discussion section, the authors reflect on research from bimanual movement control to identify potential accounts for the bilateral encoding architecture. However, the literature to which they refer leaves room for alternative interpretations. Perhaps, the authors may be interested in a neural model of bimanual control based on TMS interhemispheric connectivity research between PMd and M1 in which left PMd (and not right PMd) is hypothesized to be in charge of the division of labor assigned to the left and right hands during the preparation of bimanual movement (Fujiyama et al., 2016a,b). The latter bimanual work is generally consistent with the hemispheric asymmetry evidence provided in the present manuscript but also reflects on the specific function assigned to the left premotor cortex during bimanual control. This suggests potentially alternative reasons for the finding that electrodes that generalized well across arms were primarily located in premotor regions of the left hemisphere.

Thanks for pointing out this relevant reference for our discussion on bimanual control. The text has been updated in the Discussion section (page 21, lines 704-707).

Reviewer #2 (Recommendations for the authors):Personally, I feel the term 'bilateral kinematic encoding' (or 'bilateral encoding'), as it appears in the title or text, is slightly ambiguous, as the 'dominance' for 'bilateral kinematic encoding' can imply that both the contra- and ipsilateral encoding were stronger than the other hemisphere. However, the contralateral kinematic encoding was not explicitly tested in the current paper. It is entirely up to the authors, but I think 'ipsilateral kinematic encoding' would be less confusing than 'bilateral kinematic encoding.' The authors could otherwise compare the contralateral kinematic encoding between the hemispheres.

We choose the term bilateral instead of ipsilateral because we wanted to emphasize the fact that the left hemisphere appears to be encoding movements produced by both the ipsilateral and contralateral arm.

It was not clear from the text whether the electrodes were pooled over the three patients for each hemisphere group or the patients were still treated as a random effect. Please clarify this point. In either case, it would be informative to show the breakdown of these electrodes in terms of patients (e.g., page 5, line 25 "665" electrodes, page 7, line 11, "141", "75"). A summary table of the electrodes indicating the participants and regions they belong to would be very helpful for readers. As the co-registration process of electrodes on the reconstructed cortical surface indicated by Stolk et al. (2018) uses the freesurfer, it would be relatively easy to find in which of the freesurfer-defined regions, such as Destrieux Atlas (Destrieux et al., Neuroimage, 2010), each of the electrodes covered. Please consider.

Please see our response to Essential Revisions #1 above where we discuss at length the issue of electrode variability and how we had addressed this in the revised manuscript. In brief, we now treat participant as a random effect in the linear mixed effects model. We have also added a new supplementary, S2, which has a summary of electrode locations classified by regions using the Desikan-Killiany Atlas. While the Destrieux Altas has more parcellations, many are within the sulci and thus are not well reflected in activity of the ECoG grids; as such, the Desikan-Killiany Atlas provides similar information and is easier to visualize given it contains fewer parcellations.

Please consider using different colors or marker types for different patients for the scatter plots displayed in Figures 2-5, as this is more informative to readers.

Thanks for the suggestion. Figures 2-5 now use different colors for each participant.

A recent line of studies (Ariani and Diedrichsen, J Neurophysiol, 2019; Ariani et al., J Neurophysiol, 2020) emphasizes the significant contribution of online planning in the production of sequential movement. In this context, the even stronger bilateral shared encoding observed during the movement phase might partially reflect the online planning of return movements. As this return movement is always to the same starting position (target), higher similarity could be explained in part by this component. Please discuss.

This is an interesting idea and we now address this in the Discussion (Page 20, lines 674-684).

Please specify the reason why the authors excluded the peripheral targets for the cross-arm generalization analysis. These targets could provide further insight into under which coordinate frame (i.e., extrinsic or intrinsic) the contra- and ipsilateral encoding overlapped (e.g., Wiestler et al., J Neurosci, 2014; Haar et al., J Neurosci, 2017).

Only the two center targets (2,4) were common to the left and right arm conditions. The two peripheral targets (1,3) were in different locations and thus do not make a good comparison. We opted to prioritize the number of trials per target at a cost of having a larger target space given the limited testing time for each participant.

Although up to the authors, it would be recommended to report "noise ceiling" for each electrode (e.g., Schoppe et al., Front Comput Neurosci, 2016). The noise ceiling would also help interpret whether poor prediction accuracy is due to the low SN ratio or the inappropriate model in the across-arm generalization test.

We have added a new Supplemental Figure, S3, to show the calculations for CCnorm (recommended by Schoppe et al., 2016) alongside our primary encoding metric (CCabs; average held-out correlation between the actual HFB and the predictions based on the model). CCnorm is normalized by signal power whereas our primary metric – Pearson’s product-moment correlation coefficient is not. We find that CCnorm does not drastically differ from CCabs, suggesting that we have sufficient trials and signal power in our dataset such that the encoding model does a good job of capturing the majority of explainable variance. The text has been updated in the Method section (Page 26-27, lines 956-972).

With respect to our interpretation of electrodes that fail to generalize, we have also added a new panel to Figure 5 (Figure 5B) which is discussed in more detail in the next comment.

Figure 5B: It would be more informative also to show all other electrodes (or areas covered by each electrode grid, for instance, by colored rectangles) to indicate the coverage for each hemisphere group.Similarly, displaying the simple ipsilateral encoding strength (shown in Figure 2) in the same format of Figure 5B (categorized by generalization index) would also be informative for comparing the shared and unshared ipsilateral encoding.

We have added an outline to the figure 5D (which used to be 5B in the previous version) to detail the extent of coverage across participants in each hemisphere. We considered outlining the coverage for each patient individually, but thought it made the figure too busy.

We have added a new panel to Figure 5 (5B in revised figure) to summarize ipsilateral encoding strength for electrodes that generalize well and electrodes that generalize poorly. As can be seen in the figure, ipsilateral encoding strength is lower for electrodes that generalize poorly, a point evident in Figure 5C (ipsilateral signals have lower amplitude).

Beyond the difference in ipsilateral encoding, in Figure 5C we can examine the temporal structure of the two signals. For electrodes that generalize poorly there is a divergence in temporal structure prior to movement onset, captured by the negative correlation between the HFA signal encoding ipsilateral or contralateral movements. Thus, electrodes that generalize poorly have lower ipsilateral signal strength and differ in terms of the temporal pattern for ipsilateral and contralateral movement.

We have revised the text and Figure 5 to clarify our arguments on this issue (pages 12-14, Results section: Temporal and spatial topography of across-arm generalization). This led us to switch the order so that we now first discuss the temporal relationship and then the spatial relationship.

Schaffer et al. (2020) was not found in the reference list (Schaffer and Sainburg, 2021 was listed instead). Please list the proper reference.

Thank you – we have updated the reference list.

Please report the age data of the participants in the Methods.

Patient ages have been added to Table 1.

Please provide more details about the statistical analyses in the Methods.Page 20, line 1: Please specify what the MD, n, i, and x indicate.Page 20, line 5: Please specify what the SSEe, n, i, contra_i_, and ipsi_i_, indicate.Page 20, line 12: "min" is omitted before "(SSEe)" in the denominator.

We have clarified variable names and indices and provided more details about our statistical analyses in the Methods section.

Reviewer #3 (Recommendations for the authors):1. As stated above, I believe that the recording sites between hemispheres must be matched; this is the only way to ensure that the inter-hemispheric differences reported are not simply due to the intra-hemispheric differences reported in Supplementary Figures 1-2.

We agree: Please see our response to Essential Revisions #1 above where we discuss at length the issue of electrode variability and how we had addressed this in the revised manuscript. In brief, we have added a new figure (S2), quantifying the number of electrodes in each subregion.

2. I'm a bit confused as to why the landmark used in Supplementary Figures 1-2 is the dorsal aspect of the central sulcus. I might expect that premotor electrodes would generalize better (and exhibit more bilateral encoding) than M1 electrodes, but your analysis seems to make the argument that face M1 electrodes also generalize better than arm M1 electrodes. Could you provide a bit more interpretation of this empirical result?

Part of the motivation here was empirical: We observed that only electrodes located close to the arm area of motor cortex failed to generalize across arms. A post-hoc explanation would be that electrodes over motor cortex are more closely tied to actual arm movement. However, we recognize that other areas (including within M1) might be encoding information about the contralateral arm, even if that information is more abstract (e.g., about the movement goal). Indeed, signals from microelectrode arrays in the human hand knob have been shown to be related to movements performed by the face, head, arm and leg (Willet et al., 2019), suggesting (yet again) the need to reconsider the notion of the motor homunculus. Similarly, we see encoding (strong levels of prediction performance) in other areas of M1 that are likely distant from the arm area of M1. For example, in more ventral aspects (corresponding to face area in the conventional maps), we see more similar encoding across the two arms suggesting that this information is more abstract.

3. I don't quite get the relevance of Churchland 2012 results to the trace in Figure 5C. Churchland et al. used only outward reaches in the 2012 study, whereas your subjects are performing out-and-back movements. The trace in Figure 5C is more similar to the trigger signal reported in Kaufman, et al. In fact, in monkeys making movement sequences (comparable to your out-and-back movements), Zimnik and Churchland (2021) found a biphasic trigger signal that looks pretty similar to the trace shown in 5C. To that end, it might be useful to plot the across-subject hand speeds in 5C. If the ECoG signal plotted is related to the trigger signal, then it should have a time-course similar to the subjects' hand speed.

Upon reflection, we agree that this reference was inappropriate. We have revised the Results section by citing two papers that are more relevant, Kaufman et al. (2016) and Zimnik and Churchland (2021). In addition, we have added a new section and figure (Figure 7) to the Results section to address whether the HFA is related to the condition-invariant trigger signal. Specifically, Figure 7 compares HFA to averaged kinematic activity for electrodes that generalize well or poorly. Two results stand out from this figure, the first is that for electrodes that generalize poorly (primarily located over M1), the HFA signal resembles speed, suggesting that this activity is related to the condition-invariant trigger signal. We find this relationship for both contralateral and ipsilateral reaches, despite the decreased amplitude of the HFA during ipsilateral movements. Second, for electrodes that generalize well we find that the HFA most closely resembles position. We note that the kinematics lag behind HFA by approximately ~200 ms for both speed and position, in line with Kaufman et al., (2016, see their Figure 6). This new section and figure are discussed in the Results section (page 16 lines: 523-537).

4. Related to 5C, wouldn't it make more sense to plot the actual HFA amplitude, rather than the predicted HFA amplitude? I understand that the two will likely be very similar, but 5C is making a point about the true neural activity.

Thanks for pointing this out. We have changed these panel from predictions to actual HFA amplitude. As the reviewer predicted, the two look quite similar but we agree it makes more sense to plot HFA given our aim is to interpret neural signals.

[Editors' note: further revisions were suggested prior to acceptance, as described below.]

The manuscript has been improved but there are some remaining issues that need to be addressed, as outlined below:1. All reviewers were very happy with the revisions and think the paper is strong. Please see the remaining comments of Reviewer #2 especially with respect to statistical support for some of the claims.Reviewer #2 (Recommendations for the authors):The authors adequately addressed most of my previous concerns. I have one further recommendation and a comment related to the modification.The authors need some statistical support for their claims regarding (i) hemispheric equality in regional distribution of electrodes (lines 226-228) and (ii) hemispheric difference in the number of electrodes with good/poor across-arm generalization (lines 412-414, 439-441, and 590-597).(i) Hemispheric equality in regional distribution of electrodes (lines 226-228):"When the categorization data were pooled across the three left hemisphere and three right hemisphere patients, the proportion of electrodes was similar in premotor, sensorimotor motor, and parietal regions." (Lines 226-228)This can be tested by using, for example, multinomial goodness-of-fit tests using expected frequency for each region estimated from the data from either hemisphere. Reporting the number or proportion (%) for these three regions in the main text, in addition to the bar graphs, would also be helpful.

We now report a chi-squared goodness of fit test, assessing if the proportion of electrodes across the eight parcellations were different for the left and right hemisphere patients. We have added the following text to the paper:

‘When the categorization data were pooled for the three left hemisphere and the three right hemisphere patients, the proportion of electrodes was similar across the eight parcellations that encompass premotor, sensorimotor motor, and parietal regions [averages: premotorright= 11%,premotorleft=10%,sensorimotorright = 17%, sensorimotorright=16%,parietalright=5%,parietalleft=3%;(7) = 0.057, *p* =.999; Figure 2—figure supplement 1].’

(ii) Hemispheric difference in number of electrodes with good/poor across-arm generalization (lines 412-414, 439-441, and 590-597):"This categorization scheme revealed a striking anatomical division, with electrodes showing good across-arm generalization clustering in the left premotor and superior parietal regions and electrodes that generalized poorly clustering in left and right M1. This result does not appear to reflect a sampling bias. There are more left hemisphere electrodes compared to right hemisphere electrodes because two of the left hemisphere patients had high density grid implants. However, the electrodes in our left and right hemisphere samples were similarly distributed over premotor, sensorimotor and parietal regions (see Supplementary Figure 2)." (Lines 590-597)The fact that the electrodes in the left and right hemispheres have similar distribution over these regions does not automatically protect from the risk of finding more electrodes in the left hemisphere due to the hemispheric bias in electrode number, and need formal statistical tests. Similarly to the first point above, this can be tested by using goodness-of-fit tests (e.g., chi-squared test, likelihood-ratio G test, etc.) using the expected number of electrodes that should be found in the same region in the other hemisphere under the null-hypothesis of hemispheric symmetry (e.g., if 20 electrodes are found in the left hemisphere, 74*20/141 electrodes should also be found in the right).

Similar to above, we performed a chi-squared goodness of fit test to assess if the left and right hemispheres have similar number of electrodes that generalize well and generalize poorly. We have added the following to the paper:

‘We next examined the relationship between generalization, hemisphere and spatial position on the cortical surface. […] In contrast, electrodes showing poor generalization were observed in both hemispheres (proportion of electrodes that generalize poorly: L1 = 21%, L2 = 5%, L3 = 6%, R1 = 14%, R2 = 15%, R3 = 37%; χ2(2) = 0.400, *p*=.818).’